# Highly parallelized laboratory evolution of wine yeasts for enhanced metabolic phenotypes

Payam Ghiaci[1,2,3], Paula Jouhten [iD][3,4,5], Nikolay Martyushenko[6], Helena Roca-Mesa[7], Jennifer Vázquez[7,8], Dimitrios Konstantinidis[3], Simon Stenberg[1], Sergej Andrejev[3], Kristina Grkovska [iD][3], Albert Mas [iD][7], Gemma Beltran[7], Eivind Almaas [iD][6✉], Kiran R Patil [iD][3,9✉] & Jonas Warringer [iD][1✉]

## Abstract

**Adaptive Laboratory Evolution (ALE) of microorganisms can improve the efficiency of sustainable industrial processes important to the global economy. However, stochasticity and genetic background effects often lead to suboptimal outcomes during laboratory evolution. Here we report an ALE platform to circumvent these shortcomings through parallelized clonal evolution at an unprecedented scale. Using this platform, we evolved $10^4$ yeast populations in parallel from many strains for eight desired wine fermentation-related traits. Expansions of both ALE replicates and lineage numbers broadened the evolutionary search spectrum leading to improved wine yeasts unencumbered by unwanted side effects. At the genomic level, evolutionary gains in metabolic characteristics often coincided with distinct chromosome amplifications and the emergence of side-effect syndromes that were characteristic of each selection niche. Several high-performing ALE strains exhibited desired wine fermentation kinetics when tested in larger liquid cultures, supporting their suitability for application. More broadly, our high-throughput ALE platform opens opportunities for rapid optimization of microbes which otherwise could take many years to accomplish.**

**Keywords** Experimental Evolution; Evolutionary Engineering; Yeast; Metabolism; Fermentation
**Subject Categories** Biotechnology & Synthetic Biology; Evolution & Ecology; Microbiology, Virology & Host Pathogen Interaction

## Introduction

Microbial processes play important roles in the global economy with the production of fermented food and drinks alone accounting for trillions of dollars in turnover (Bisson et al, 2002; Jullesson et al, 2015; Marsit and Dequin, 2015). Beyond fermented food and drinks, microbial fermentation is central to many production processes, such as enzymes, antibiotics, probiotics, fine chemicals, and biofuels. However, the naturally occurring microbes seldom operate at the efficiencies required for an economically viable production, leading to intense efforts to improve their properties (Dai and Nielsen, 2015; Dugar and Stephanopoulos, 2011; Peris et al, 2018). Rational genetic engineering has helped in addressing some microbial shortcomings (Choi et al, 2019; Nielsen et al, 2014; Oud et al, 2012), but generates genetically modified organisms (GMO) that are often unsuitable or restricted for food, feed, or beverage production. Further, successful genetic engineering requires a deep understanding of the genotype–phenotype map (Brochado and Patil, 2013; Lee and Kim, 2015; Monk et al, 2016; Steensels et al, 2019), which is often unknown, especially in the cases of complex, multi-genic traits. Mathematical modeling of cellular states and fluxes is equally challenging due to the interconnected nature of metabolic and regulatory processes (Basler et al, 2016). Manifestation of both the desired industrial trait and unwanted side effects therefore often depends both on the genetic background and the environment (Costanzo et al, 2016; Stearns, 2010; Streisfeld and Rausher, 2011) and defy prediction.

Adaptive Laboratory Evolution (ALE) offers an attractive alternative for microbial improvement because it is unburdened by the need to understand the genotype–phenotype relation on a single gene level (Mans et al, 2018; Notebaart et al, 2018). As such, it has been successfully used for, e.g., microbial thermotolerance (Caspeta et al, 2014), methylotrophy (Espinosa et al, 2020), carotenoids production (Reyes et al, 2014), and alcohol tolerance (Ghiaci et al, 2013; Wang et al, 2020). Yet, ALE lineages often fail to evolve the desired traits, or end up carrying unwanted side effects, because the etiology of the desired traits involves neutral, costly, inaccessible, or highly pleiotropic mutations (de Visser and Krug, 2014) (Fig. 1A). Moreover, chance influences both the birth and early fate of mutations, delaying the establishment of beneficial variants in ALE populations and allowing neutral or weakly deleterious variants to become common (Masel, 2011) (Fig. 1B). The power of numbers could unshackle ALE from the constraints

[1]Department of Chemistry and Molecular Biology, University of Gothenburg, PO Box 462, Gothenburg 40530, Sweden. [2]Department of Biorefinery and Energy, High-throughput Centre, Research Institutes of Sweden, Örnsköldsvik 89250, Sweden. [3]European Molecular Biology Laboratory, Heidelberg 69117, Germany. [4]VTT Technical Research Centre of Finland Ltd, Espoo 02044 VTT, Finland. [5]Aalto University, Department of Bioproducts and Biosystems, Espoo 02150, Finland. [6]Department of Biotechnology and Food Science, NTNU - Norwegian University of Science and Technology, Trondheim, Norway. [7]Universitat Rovira i Virgili, Dept. Bioquímica i Biotecnologia, Facultat d'Enologia, Tarragona 43007, Spain. [8]Centro Tecnológico del Vino—VITEC, Carretera de Porrera Km. 1, Falset 43730, Spain. [9]Medical Research Council (MRC) Toxicology Unit, University of Cambridge, Cambridge CB2 1QR, UK. ✉E-mail: eivind.almaas@ntnu.no; kp533@cam.ac.uk; jonas.warringer@cmb.gu.se

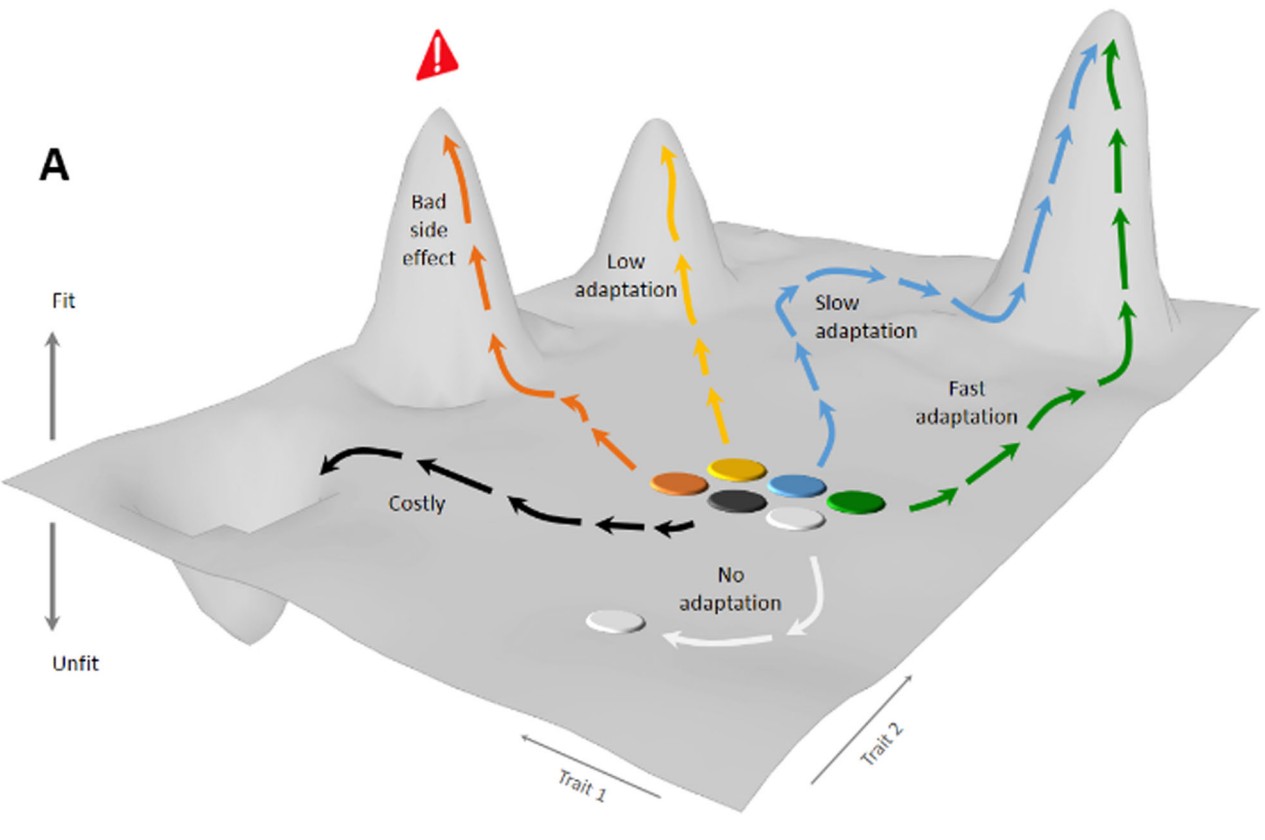

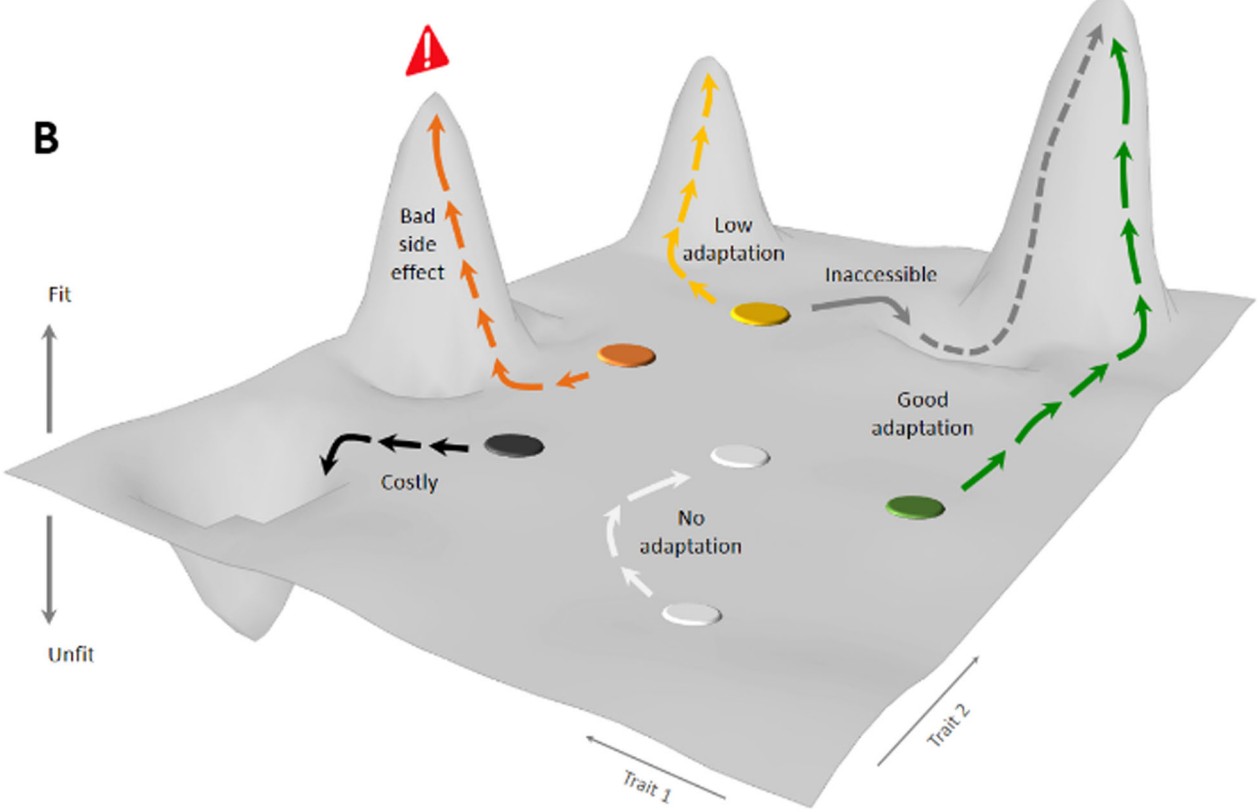

**Figure 1. ALE challenges overcome by parallelization.**

(A) Mutations underlying desired traits may be neutral (blue), costly (black), inaccessible (gray), low adaptive (yellow), or associated with side effects (orange) in a lineage. Parallelizing ALE across many lineages can increase the chances of some being capable of evolving a desired trait that is unburdened by side effects (green). (B) Chance affects new mutations in a population, leading to slow (blue), or low yield (yellow), adaptation, and a side-effect burden (orange). Parallelizing ALE across many replicated populations of a lineage can increase the chances of some quickly adapting to a high adaptation yield without becoming burdened by unwanted side effects (green).

of chance; however, achieving a sufficiently high ALE throughput without compromising error rates has proven challenging (Dunham et al, 2017; Fasanello et al, 2020; Lukačišinová et al, 2020; Nguyen Ba et al, 2019; Wong et al, 2018). We here developed an ALE platform capable of evolving $10^4$ microbial populations in parallel, and we demonstrate the utility of parallelization by selecting eight desired wine production traits while measuring adaptation, and its side effects, with high accuracy.

## Results

### Parallelizing ALE for high throughput

We developed a highly parallelized ALE platform and evaluated its applicability by evolving 48 wine yeasts for eight desired wine yeast traits (Appendix Table S1). The selected traits include growth on less-preferred nutrients (fructose and arginine, proline), growth on eccentric concentrations (high sugar, high ethanol, low vitamin and low nitrogen) and production of metabolites (glutathione and aromatic compounds) toward a more efficient and better-quality wine production for which we designed specific environments (Appendix Tables S2 and S3). We used 15 commercial wine yeasts (Lallemand Inc. Montreal, Canada) and 33 noncommercial lineages isolated from cellars across the Priorat wine district in Spain as genetic starting points and sequenced their genomes. Commercial and noncommercial lineages all shared recent ancestry (mean nucleotide diversity = 0.000018–0.011) within the Wine/European clade and showed only limited population structure (Liti et al, 2009; Peter et al, 2018) (Fig. EV1). Heterozygosity varied across strains (heterozygote/homozygote sites = 0.02–0.67) with half of strains (55%) being almost completely homozygotic (heterozygosity ratio <0.05) (Appendix Table S1). We repeatedly ($n = 24$ replicate ALE populations) expanded all 48 lineages clonally to generate 1152 parent strain populations. We cultivated these as colonies on solid medium based on each of the eight synthetic grape must media (Beltran et al, 2004) ($n = 9216$), as shown in Appendix Fig. S1A,B.

To estimate doubling times for the parent population at the start of the ALE regime, we used an automated set-up (Zackrisson et al, 2016) to count cells in colonies expanding on the designed variations of a solidified synthetic grape must, achieving generally high precision (mean CV = 10.0%) (Appendix Fig. S1C). Selection environments increased the cell doubling time (mean increase across strains and environments: 4.15 h or 200%) compared to control environments; thus, they imposed substantial selection pressures (Fig. EV2A, right panel). The exception was nitrogen starvation, which imposed little to no selection for improvement of the maximum growth rate, likely because nitrogen supplies have yet to become limiting when growth is at its fastest. However, the

variation in cell doubling times across lineages was large, reflecting that the degree of selection for doubling-time improvement varied considerably (Fig. EV2A, heatmap). Commercial lineages enjoyed no cell doubling-time advantage over noncommercial in any niche (Fig. EV2A,B), and a clustering of strains based on phenotypic similarity often grouped commercial lineages together with noncommercial lineages, with no evident connection to the weak population structure (Fig. EV2A, heatmap). However, many lineages in both categories were general slow growers (Fig. EV2A, bottom panel), which imply no or limited historical adaptation to the background grape must medium.

We evolved the 9216 wine yeast populations over 30 consecutive ALE batch cycles and stored evolved endpoint strains as frozen stocks (Appendix Fig. S1B). Extinctions of 2351 populations (~25%) mostly affected slower-growing lineages and likely reflected ratchet-like error accumulation, and fewer viable cells being transferred to new batch cycles, as the ALE progressed (Fig. EV3A). Surviving ALE populations varied dramatically in their endpoint doubling-time adaptation, and differences between selection environments could explain 45% of this variance (Fig. 2A,B). This reflects that adaptability in some environments is constrained across wine yeasts, while being generally high in others. Indeed, nitrogen starvation and selection for better aroma production often reduced, rather than improved, fitness (population doubling times). This is consistent with the high rates of extinction in these two designed niches, whereas the other six environments tended to promote fitness gains, albeit to varying degrees. Adaptation was strongest in high-sugar grape must, in which 30% of the populations achieved >80% doubling-time reductions. The capacity of wine yeasts to rapidly improve when facing environments richer in sugar than typical grape musts (200–250 g/L) suggest limited historical selection to perform well in such niches.

### Highly parallel ALE enables the selection of better-adapted variants

General differences between wine yeasts explained 23% of adaptation variance, illustrated by the strain E11 often adapting fast (mean doubling-time reduction: 1.15 h) while strain T7 consistently reached (Fig. EV3A) or approached extinction (Fig. 3A,B). Wine yeast adaptability therefore has a generic component that persists across selection environments, with some strains being inherently more suited to ALE in synthetic grape must. The commercial and noncommercial lineages were equivalent in this respect (Fig. EV3C). However, more often than not, adaptability involved significant genotype-by-environment components (Fig. EV3D), as illustrated by T14 and D10 showing approximately the same overall ability to adapt to the selection pressures imposed, but D10 adapting much better to fructose use and T14 much better to vitamin scarcity (Fig. 3A). Predicting wine

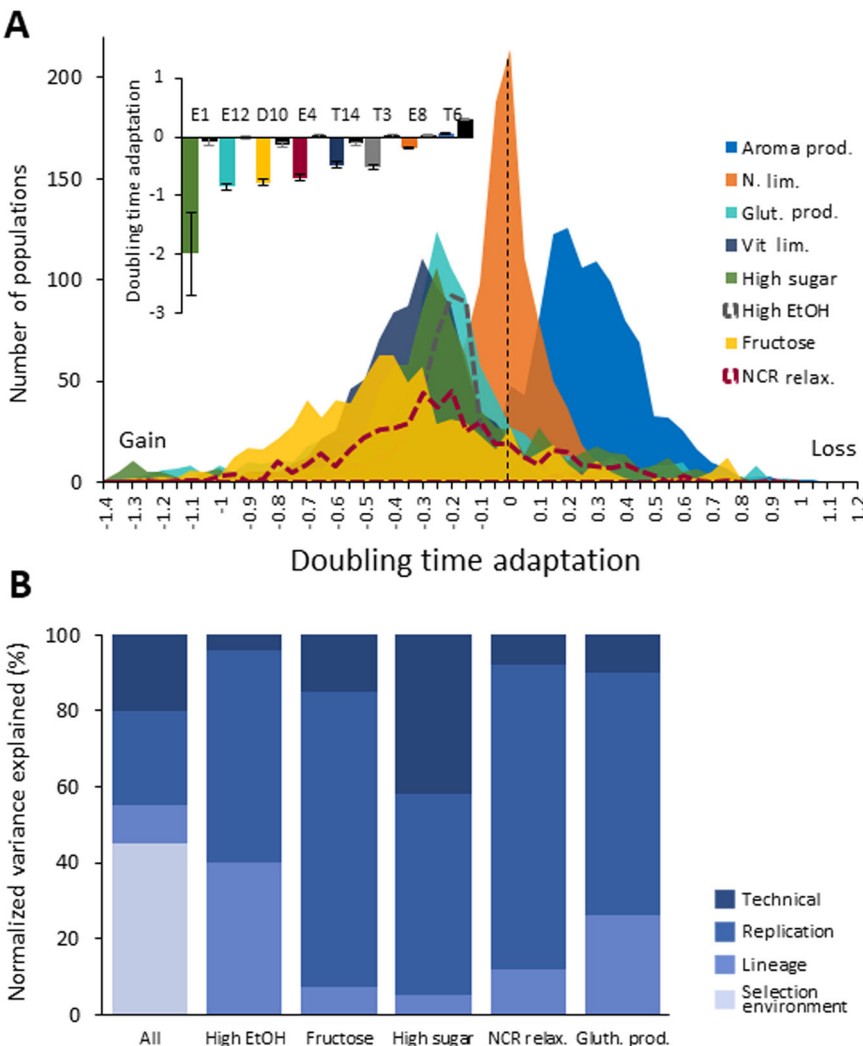

**Figure 2. Wine yeast adaptation under highly parallelized ALE.**

We parallelized ALE across many ($n = 24$) replicated populations of 48 (Appendix Table S1) commercial and noncommercial wine yeasts as replicated colony populations on each of eight synthetic grape musts designed to select for traits desired by the wine industry (Appendix Tables S2 and S3). The ALE design for the 9216 populations is shown in Appendix Fig. S1. We counted cells in growing populations to generate high-density growth curves and estimated adaptation as the log(2) change in normalized cell doubling time from before to after ALE. (A) Histogram of the adaptation of all ALE populations, in each ALE environment (color). Extinct ALE populations are not included. Inset: Mean ($n = 24$ replicate ALE populations) adaptation of the wine yeast lineage with the greatest adaptation (left bar, indicated by name) as compared to the average wine yeast lineage ($n = 48$ strains), in each selection environment. Error bars: SEM. (B) Percentage of the variance in adaptation that is explained by differences between: ALE selection environments, parental lineages, replicated ALE populations, and technical replicates. Source data are available online for this figure.

yeast adaptation in a particular ALE environment is therefore likely to be challenging, and this is further underscored by the often-best adaptation predictor, low initial fitness (Jerison et al, 2017; Persson et al, 2022; Stenberg et al, 2022), accounting for no more than 36% of adaptation variance (Pearson, $r = -0.61$) (Fig. 3B, left panel). An unbiased high-throughput ALE platform based on the power of numbers circumvents this need for predictability by allowing a simple post-experiment selection of the most improved strains for further development. The value of such an approach is illustrated by the fact that the most adapted lineages reduced their doubling time 1.1–3.8× more than the average lineage in each environment (Fig. 3C, main panel) and often ended up among the fastest growing (Fig. 3B, right panel). The latter underscores that ALE not only quenches defects in low-fitness lineages, transforming these

into average wine yeasts, but can imbue them with a fitness that is much superior to what is typical. This is illustrated by the gains of D10 in an environment selecting for fructose, and of E11 when selecting for NCR (Nitrogen Catabolite Repression) relaxation (Fig. 3C, main panel).

The divergence of populations evolving as replicates further amplified the value of ALE parallelization, accounting for nearly 70% of adaptation variance within environments (Fig. 2B). The most adapted ALE replicate of each parental lineage (1.1–1.9×) consistently reduced its cell doubling time more than the average ALE replicate for that parental lineage (FDR: $q = 0.05$). For the best-adapting strains (average of replicates) in each environment, this advantage of the best adapted replicate was even larger (1.1–6.6× greater doubling-time reduction than the mean replicate)

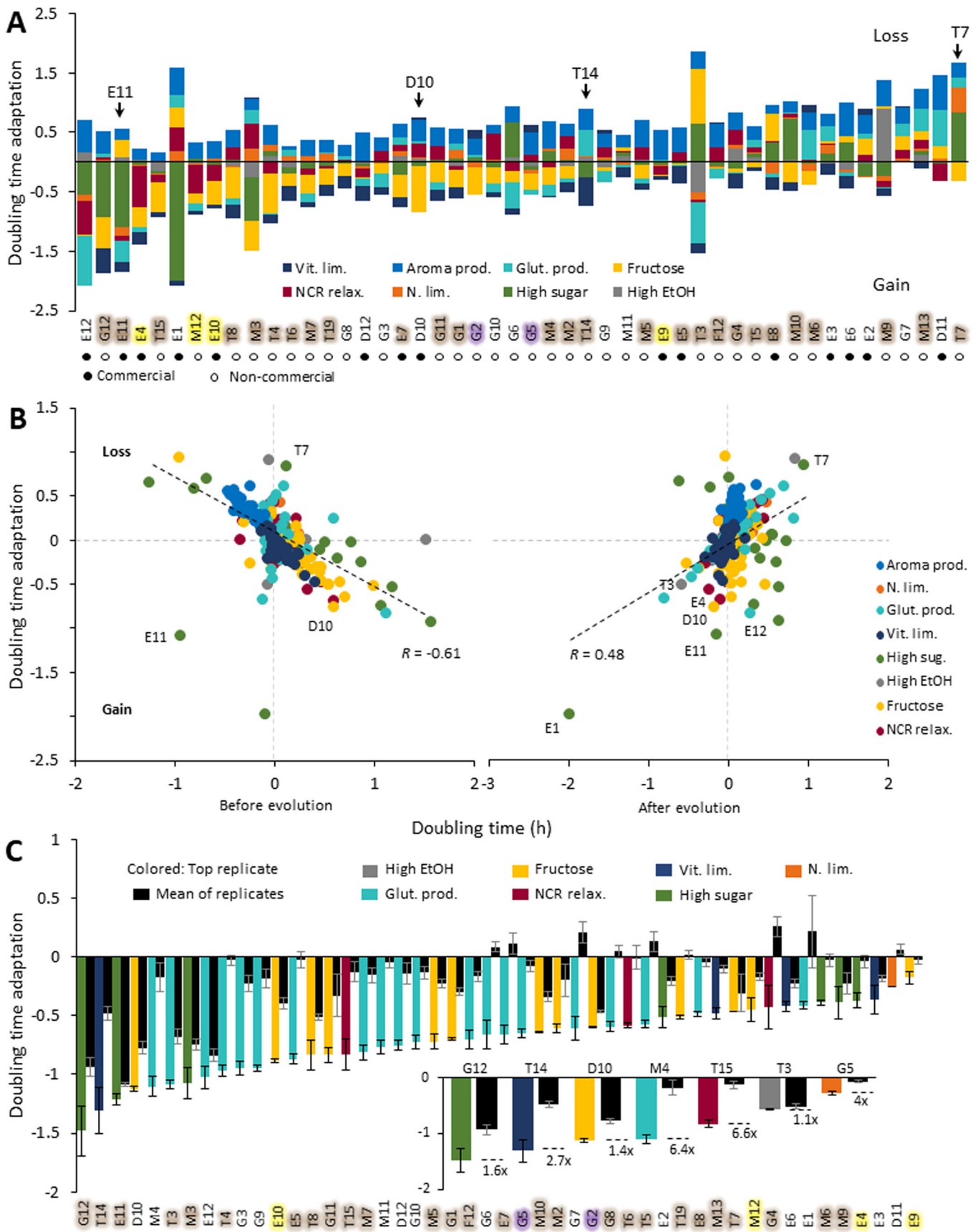

(A) Stacked bar plot of the mean ($n = 24$ replicated ALE populations; extinct populations excluded) adaptation for each parental wine lineage, in each ALE selection environment. ALE selection environments with positive adaptation (cell doubling-time decrease) are shown as negative sums, environments with negative adaptation (cell doubling-time increase) are shown as positive sums. Zero: no adaptation. Text color: population (see Fig. EV1). Arrows: lineages indicated in text. (B) Comparing mean ($n = 24$ replicated ALE populations) adaptation, for each parental wine yeast in each ALE selection environment, to the mean relative doubling time before (left panel) and after (right panel) ALE selection. Color = selection environment. Broken line = linear regression. Pearson's $r$ and lineages mentioned in text are indicated. (C) Adaptation (mean, $n = 4$ biological replicates) of the most adapted replicated wine yeast lineage (left bar, indicated by name) as compared to the average lineage ($n = 24$ replicated ALE populations) across selection regimes. Inset: adaptation (mean, $n = 4$ biological replicates) of the top replicate of the most adapted wine yeast lineage (left bar, indicated by name) as compared to the average lineage ($n = 24$ replicated ALE populations) in each selection regime. FDR: $q = 0.05$. Error bars: SEM. Source data are available online for this figure.

(Fig. 3C, inset panel, and Appendix Fig. S2). These benefits of ALE replication depended strongly on selection environment and genetic background, with selection for a relaxed NCR, tolerance to high sugar, and glutathione production, and lineages E12, G12, and M2 promoting a particularly large variation in adaptation between replicates (Fig. EV3D). Overall, we conclude that expanding the numbers of ALE lineages and replicates, while keeping other parameters constant, improves the chances of obtaining highly adapted variants.

## Parallelizing ALE empowers selecting against unwanted side effects

Pleiotropy, genetic linkage and neutral genetic drift means that ALE strains typically acquire traits other than those desired, often limiting their industrial usefulness. We measured side effects by cultivating the start and endpoints of 5760 evolved populations in 18 carbon or nitrogen-limited environments and estimated the doubling-time change that adaptation had caused. We found side effects on carbon and nitrogen metabolism to be common, with the mean population significantly (Student's $t$ test, $P = 0.01$) changing its cell doubling time in 15 out of 18 carbon or nitrogen-limited environments. These side effects tended to be substantially weaker than adaptations (Fig. EV4A). Overall, they also enhanced and impaired growth equally often, but this varied across ALE environments, with e.g., selection on the NCR medium strongly tending to give negative side effects. Thus, the capacity to grow under carbon restriction, particularly when provided as galactose, mannose, or maltose, often improved despite the absence of direct selection for use of these carbon sources in the designed selection environments (Fig. 4A). In sharp contrast, the capacity to grow under nitrogen restriction, in particular when using aspartic acid, valine, and urea as sole nitrogen sources, consistently deteriorated across selection environments. This dichotomy likely reflects stronger selection on a fast carbon catabolism in the carbon-rich (20%) background medium of synthetic grape must.

Next, we grouped populations based on similarity in evolved side effects using hierarchical and t-SNE clustering and found populations adapting to the same selection environment often acquiring similar side effects (Fig. 4B,C). Thus, each selection environment allowed adapting wine yeasts to evolve a few ($n = 2$–5) distinct sets of side effects, or syndromes, with each syndrome being mostly private to one selection environment. Syndromes appeared to be universally available, as on average 96% of wine yeasts could evolve each of the sets of side effects (Fig. EV4B). However, different parental lineages had very

different propensities to evolve a particular syndrome in any given ALE niche. The strong influence of genetic background on the evolution of traits not under direct selection resulted in some wine yeasts tending to evolve more desirable sets of side effects than others, in the sense that these syndromes represented faster growth in a wide range of environments (Fig. 5). This tendency had a generic component, with e.g., M11 generally evolving fast growth side effects and G2 generally evolving slow growth side effects, regardless of ALE niche (Figs. 4B and 5). However, the parental lineages evolving the best, and the worst, sets of side effects varied substantially across ALE niches. All other factors being equal, evolving many ALE lineages in parallel therefore offers better chances of obtaining at least some ALE strains that are unburdened by unwanted side effects.

## *MEP2* and chromosomal mutations drive wine yeast adaptations to NCR relaxation

We sequenced the DNA of 26 fast-adapting populations and called single-nucleotide and small indel variants having reached substantial frequencies, but rarely fixation, relative to the corresponding parental lineage (Dataset EV1; Table EV1 and Appendix Table S6). In the case of selection for NCR relaxation (arginine, proline consumption), four out of seven populations acquired different amino acid changes in Mep2 (Fig. 6A), strongly suggesting these to have contributed to adaptation. All four *MEP2* mutations were present at frequencies of $P = 0.967$ to $0.989$, reflecting near fixation of homozygotic variants. The homozygosity likely arises from gene conversion, which is known to help drive ALE of diploid yeast populations (Vázquez-García et al, 2017a). Mep2, a high-affinity ammonium permease, serves as main entry point for the ammonium analog methylamine, which we used to activate the NCR intracellularly and to repress the use of arginine and proline, the only sources of usable nitrogen present in this ALE niche. Point mutations in *MEP2* have also been shown to prevent methylamine uptake (Wang et al, 2013a). Adaptation to this repression is therefore likely to have been driven by loss of Mep2 function and reduced methylamine uptake, which should result in NCR relaxation, and consequently faster use of the arginine and proline. One of the point mutations, W275stop, is a non-sense mutation, consistent with this explanation (Fig. 6B). An in-frame deletion of P465 in the autoinhibitory domain of the cytoplasmic tail (Boeckstaens et al, 2014) of Mep2 may result in constitutive Mep2 auto-inhibition and closure, with similar effects. We note that, while wine yeasts in ammonium-rich grape must initially take up ammonium through Mep1 and Mep3, Mep2 takes over when ammonium concentrations fall (Beltran et al, 2004). Loss of Mep2 function is therefore likely to

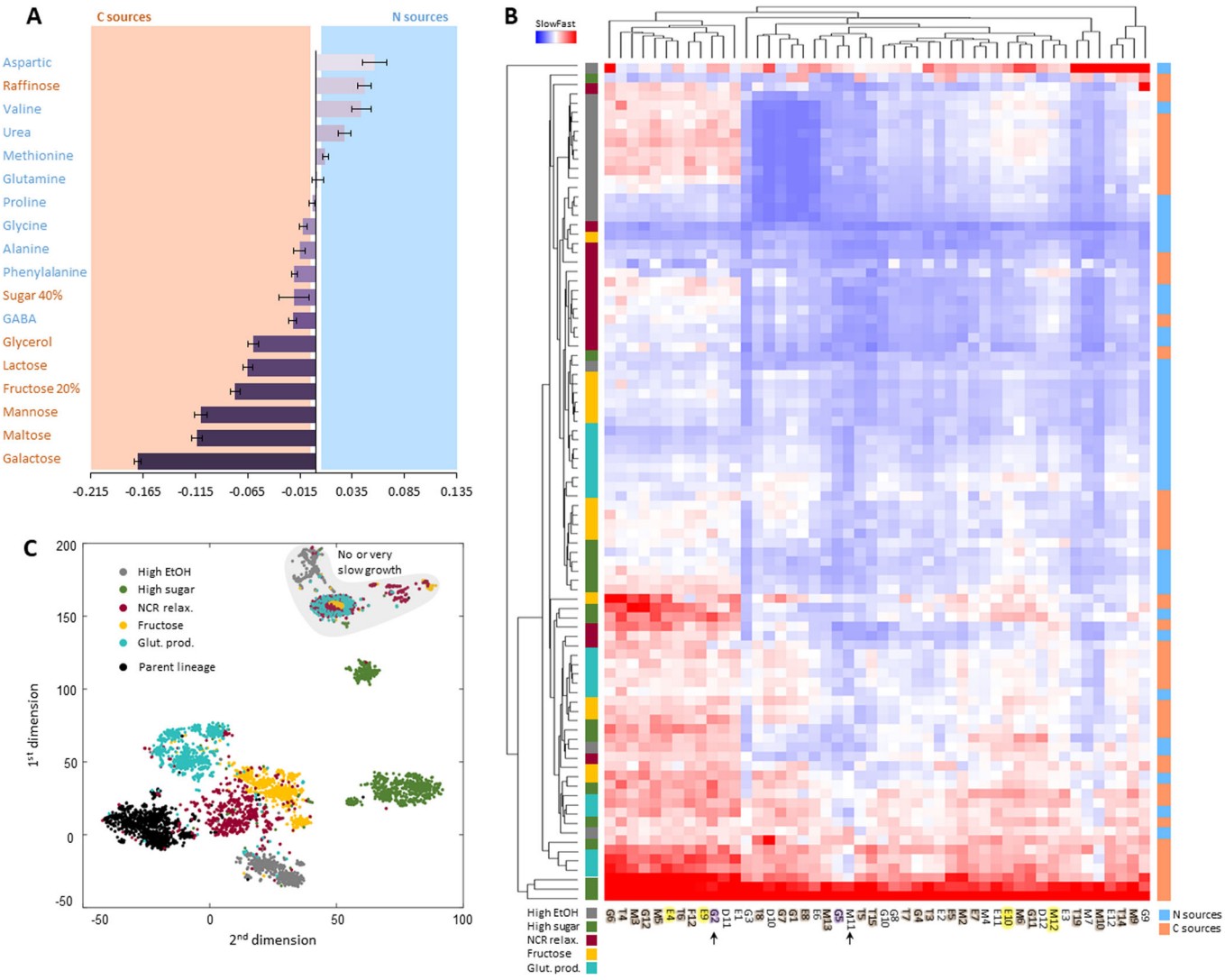

**Figure 4. Side-effect syndromes in ALE populations.**

We cultivated start and endpoints of ALE populations in eight selection environments in 18 non-selection carbon and nitrogen-limited niches. We estimated ALE side effects as the log(2) change in normalized doubling time from before to after ALE, in these niches. (**A**) Side effects of ALE, averaged across all ALE replicates, lineages and selection regimes ($n = 5760$). Faster growth side effect = negative numbers. Orange text: carbon-limited niches, blue text: nitrogen-limited niches. Error bars: SEM (across selection regimes, $n = 5$). (**B**) Central heatmap: ALE side effects evolved by each lineage in each selection regime. Each column represents one type of side effect (growth in a carbon or nitrogen-limited niche), evolved under one selection regime. Each row represents one lineage (mean of $n = 24$ replicated ALE populations), with names colored by clade (see Fig. EV1). Arrows indicate M11 and G2. Red = faster growth side effect, blue = slower growth side effect. Left panel: Hierarchical clustering of side effect niches. Left color panel: Selection regime. Note that sets of side effects are grouped by selection regime. Right color panel: Carbon (orange) and nitrogen (blue) limited niches. Upper panel: Hierarchical clustering of lineages based on similarity across side-effect-environments combinations. (**C**) t-Distributed Stochastic Neighbor Embedding (t-SNE) clustering reducing the side-effect variation to two dimensions. Each dot represents one ALE population, of one lineage, in one selection regime- or one parental lineage. Color = selection regime, as in (**B**). Note that side effects cluster by selection regime, representing common syndromes. Source data are available online for this figure.

result in slower ammonium uptake at later stages of wine fermentation, and consequently, to earlier NCR relaxation and more extensive uptake of the abundant proline and arginine, which is an industrially desired wine trait. Apart from a missense mutation in *WHI2*, encoding a TORC1 regulator, we found no genes linked to nitrogen metabolism to have been mutated in other ALE NCR relaxation populations. We also found no trace of mutations in *PUT1-4*, *GAP1*, and *URE2*, which recently were shown to drive wine yeast adaptation to a NCR relaxation medium (Walker et al, 2022).

We found few other point mutations or small insertions/deletions in ALE populations adapted to other environments, no genes that were mutated in more than one population (Dataset EV1), and no agreement with variants previously reported to drive ALE adaptation to similar selection pressures. Specifically, we found no changes to *HXT3, HXK1 or FSY1*, believed to drive fructose evolution (Berthels et al, 2008; Galeote et al, 2010; Guillaume et al, 2007), in our fructose ALE and no changes to cell wall stability genes, which have been reported to help adaptation to

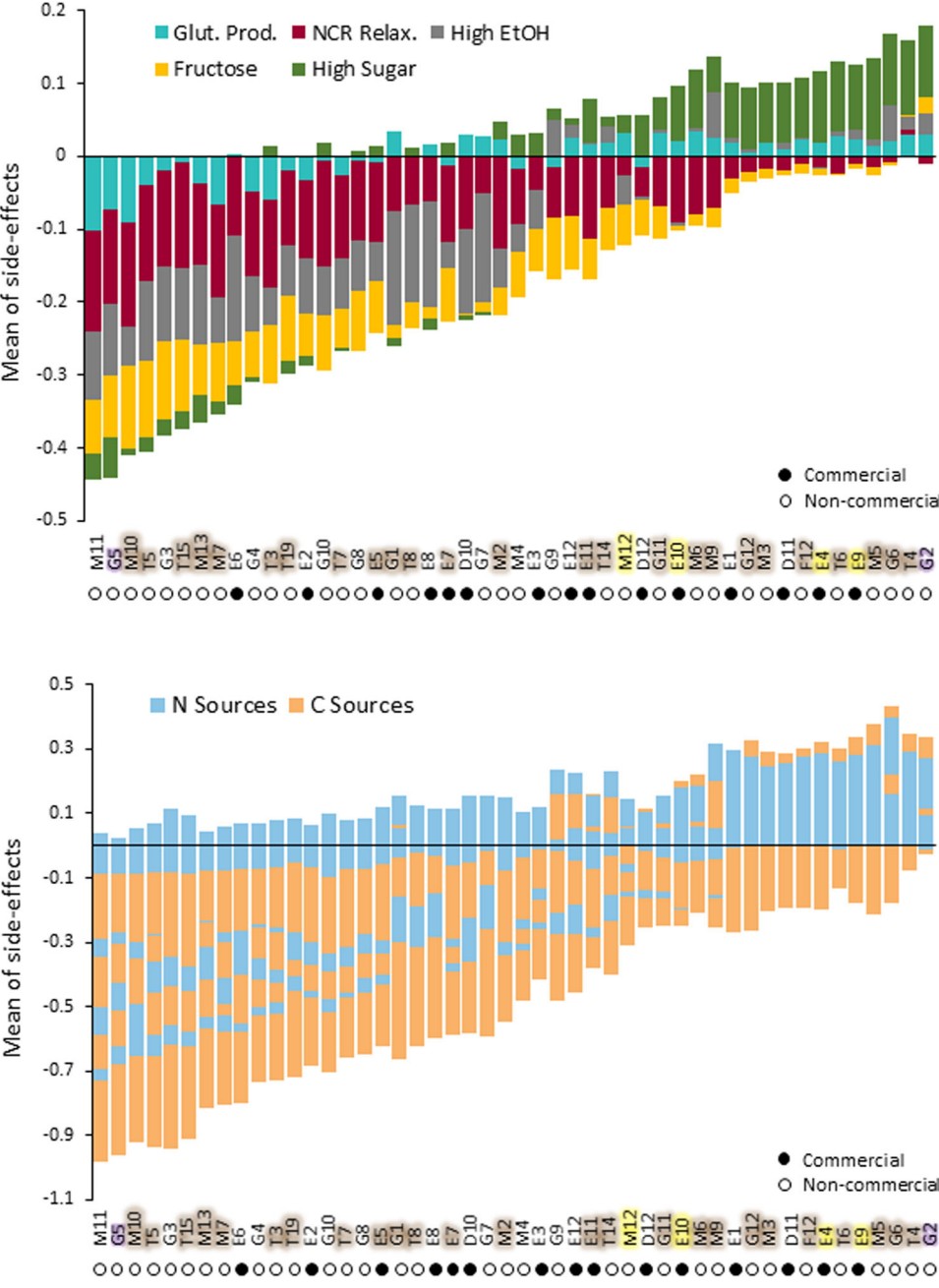

**Figure 5.  Side effects of ALE wine yeast evolution.**

Upper panel: Some wine yeast lineages (left) evolve more desirable (faster growth) side effects than others (right). Stacked bar plot of means across all side-effect environments ($n = 18$) and all ALE populations ($n = 24$) for each lineage is shown. Color = selection environment. Negative numbers: cell doubling-time reductions.
Bottom panel: Stacked bar plot of means across all ALE populations ($n = 24$) in each carbon or nitrogen environment. Strain names are colored based on population (see Fig. EV1). Source data are available online for this figure.

high alcohol concentrations (Ghiaci et al, 2013; Snoek et al, 2016), in our high ethanol ALE. Instead, all sequenced ALE populations carried large copy number variants and loss-of-heterozygosity (LOH) variants (Fig. 6C; Table EV1). Sequence coverage indicated that copy number variants, which tended to correspond to gain or loss of whole chromosome, were rarely fixed within populations, but often extensively shared across populations. Thus, 50 out of 60 such variants were present in more than one population,

consistent with them being selected for (Fig. 6C). Some, e.g., the Chr XV duplication consistently emerging in a high-sugar environment, were near private to specific ALE environments, suggesting niche-specific adaptations. High-sugar ALE has in other genetic and environmental backgrounds been linked to amplifications of chromosomes XII and IV, but not of chromosome XV (Mangado et al, 2018). Other amplifications, e.g., of Chr I and Chr XIV in high alcohol, NCR relaxation, and glutathione production

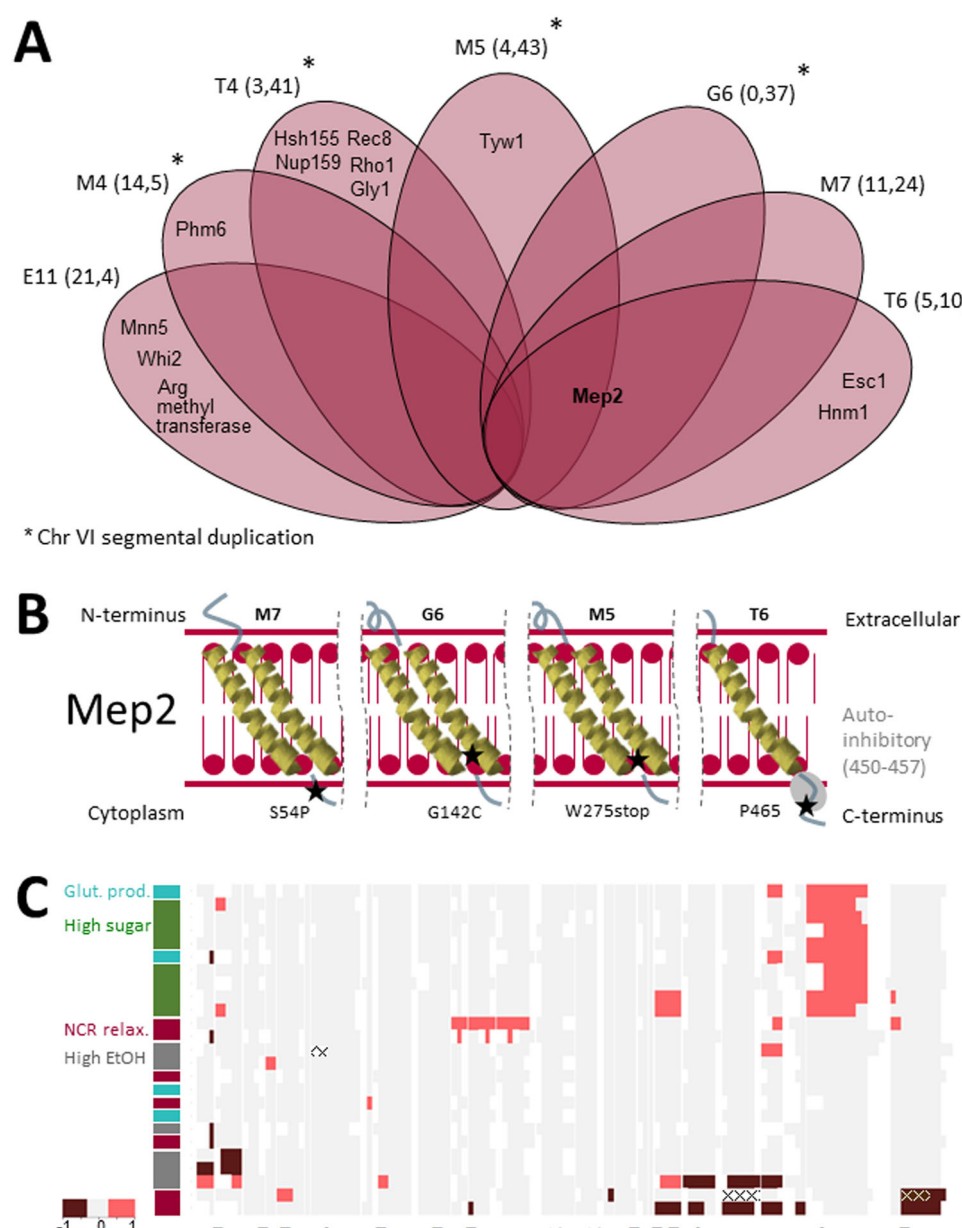

**Figure 6.** *MEP2* mutations and large copy number variants drive ALE adaptation.

(A) Venn diagram depicting proteins affected by non-synonymous mutations in seven sequenced populations ALE adapted for NCR relaxation. Names of populations (outside) and genes (inside) are indicated. Mep2 (bold) were mutated in four populations. (B) Schematic view of the high-affinity ammonium and methylammonium permease Mep2. The seven transmembrane regions (yellow), the autoinhibitory domain (gray) of the cytoplasmic tail and the amino acid mutations (asterisks) driving adaptation for NCR relaxation, and population names (top) are all indicated. (C) ALE wine yeast populations often acquire large copy number variations. Changes corresponding to 20% larger, or smaller, read coverage (central panel, color) than expected are shown for 26 sequenced ALE populations (right) evolved in four selection regimes (left panel, color). White vertical lines indicate ends of contigs. Chromosomes are indicated in roman numbers (below).

ALE niches, were shared across selection regimes. Thus, they likely reflect adaptation to the synthetic grape must background medium and potentially contributes to the good carbon use/poor nitrogen use common side-effect syndrome. Thus, although the underlying genes and mechanisms are challenging to pinpoint, many of our ALE adaptations were likely driven by large, recurring copy number variations, consistent with some earlier ALE observations (Caspeta et al, 2014; Mans et al, 2018).

## ALE populations express desired traits in larger liquid cultures

The physiology of yeast in large-volume liquid cultures differs from that of small experimental colonies expanding on top of a solid matrix (Lara et al, 2006; Neubauer and Junne, 2010; Reijenga et al, 2005; Takors, 2012). We therefore probed whether our ALE strains presented the wanted beneficial properties when shifted to larger

liquid cultures. We cherry-picked 63 endpoints from five environments with moderate to large doubling-time improvements and designed a stepwise scale-up for validation (Appendix Fig. S3A,B). We first tested whether production of the antioxidant glutathione, which protects cells against oxidative damage and can be secreted to also protect the grape must (Kritzinger et al, 2013), was present in populations adapted for this purpose, and cultivated in 40-mL liquid cultures in closed vessels. Because we could only indirectly select for glutathione production, by supplying the glutathione precursors glutamate, cysteine, and glycine as the only nitrogen sources and challenging cells with the oxidant diamine, we expected many of the populations to have adapted by other means. Nevertheless, we found two of the eighteen tested populations, M4 (15, 28) and G6 (1, 36), to substantially (27–62%, Student's $t$ test $P < 0.05$) have increased their glutathione production when cultivated in liquid (Fig. EV5A). The best performer, M4 (15, 28), produced >13 nmol glutathione/mg dry weight of biomass.

We next explored whether ALE populations selected to relax NCR, and the use of arginine and proline, even in the presence of a preferred nitrogen source, express this property when shifted to 80 mL liquid cultures. We selected 11 ALE populations having evolved faster growth on the NCR selection medium and cultivated them in liquid, mostly anaerobic cultures, using ammonium as NCR inducer. Many of these ALE populations consumed little ammonium but still grew well, consistent with NCR relaxation and concomitant fast use of arginine and/or proline (Fig. EV5B). We selected two of these populations, E11 (21, 4) and E3 (17, 32) and followed their assimilation of arginine and proline in when growing on arginine and proline as sole nitrogen sources and using methylamine as NCR inducer. Both populations fermented better than their parent strains (Fig. 7A), consistent with them being less constrained by nitrogen access. Moreover, while their proline uptake remained unchanged (Student's $t$ test, $P > 0.05$), the retention of arginine in cells improved (Fig. 7B). Yeast store arginine in the vacuole as a nitrogen reserve (Li and Kane, 2009), and if a preferred nitrogen source is encountered before this reserve has been mobilized, the arginine is instead actively exported (Opekarova and Kubin, 1997). The two NCR evolved ALE populations thus achieved their improved arginine retention by avoiding an early export of this nitrogen reserve to the environment. Our results are therefore consistent with at least some of our ALE populations reaching the desired improved arginine consumption by relaxing NCR and avoiding early arginine export to the environment.

## ALE strains perform well in grape must fermentations

We next explored whether evolved ALE populations perform well in larger cultures and in actual, rather than synthetic, grape must. For this, we selected eight ALE populations with good growth in high alcohol, high-sugar, or high fructose ALE environments. We first tested whether populations adapted to high alcohol in the form of 1.3% n-butanol, which resembles ethanol, but is less volatile and remains in the solid medium (Ghiaci et al, 2013), also grew better when exposed to high ethanol in microscale liquid cultures (Bioscreen Inc.). We found four of the six tested ALE populations to grow both faster (5–37%, Student's $t$ test $P < 0.05$) and to higher (36–137%, Student's $t$ test $P < 0.05$) cell yield in 8% ethanol than their parental lineages (Fig. EV5C). Two of these, T3 (13, 35) and

T3 (15, 35), also fermented the sugar in 50 mL liquid culture ~14% more efficiently than their parents in the absence of added ethanol (Student's $t$ test $P < 0.05$), and we selected both of these, along with E9 (0, 7) with the lowest doubling time, for wine cellar experiments (Fig. EV5D).

Next, we probed whether 18 ALE populations adapted to a high-sugar content also fermented a high-sugar synthetic grape must better when cultivated as 40 mL liquid cultures. Five populations either left less sugar in the must at the end of the fermentation than their parents or produced more ethanol per sugar molecule consumed. We retained three of these five populations for wine cellar experiments (Fig. EV5E, Student's $t$ test $P < 0.05$). Finally, we cultivated ten ALE populations adapted to fructose use, in liquid 40 mL fructose cultures also containing the glucose analog 2-deoxyglucose. We found M12 (18, 23) to grow much faster, to reach a higher cell yield, and to assimilate 35 g more fructose than its parent (Fig. EV5F, $t$ test $P < 0.05$).

We isolated and expanded single clones from the selected ALE populations and inoculated these in 27% sugar White Grenache (GR) grape must freshly harvested in DO Terra Alta (Spain). We first fermented grape must in 5 L cultures using isolates from three ALE populations adapted to high sugar and from one adapted to high ethanol adapted ALE ($n = 3$). Fermentation progressed without detectable defects in all ALE-adapted isolates, with one isolate from the high-sugar adapted population E2 (26, 37) consuming 2.4% more of the sugar than the parent strain (Fig. 7C). We examined population dynamics of this E2 (26, 37) population by comparing doubling times of 11 random single-clone isolates and the ALE population on synthetic grape must and found minute differences (Appendix Fig. S4). We accordingly selected and expanded two additional clones (iso3, iso10) and used them to ferment 80 L grape musts. Further, we isolated an extra random isolate (iso12) for a third 80 L fermentation from a different harvest. All three clones consumed the sugar fast, indicating that the good fermentative performance of E2 (26, 37) may persist also at larger scales (Fig. 7D; Appendix Fig. S5). We sequenced the genomes of iso3 and iso12 and found these to share heterozygotic missense single-nucleotide variants in the multi-functional mRNA abundance regulator Not3 and in Fau1, which helps in folic acid biosynthesis, as well as loss-of-heterozygosity segments (Appendix Fig. S6; Dataset EV1; Table EV1). The shared variants were all common in the ALE population and may have contributed to its adaptation. Our highly parallelized ALE platform was therefore capable of evolving strains that are likely to perform well also in larger scale grape must fermentations.

## Discussion

We here introduced a highly parallelized ALE platform for the improvement of industrial microbes based on expanding ALE populations over many generations as colonies on top of a designed solid selection medium and accurately counting their cells (Zackrisson et al, 2016). The parallelization allowed us to broaden the evolutionary search spectrum relative if fewer samples had been evolved by the same method, both by repeating ALE many times from a fixed genetic start point and by expanding the number of different such start points. Both expansions improved the chances of obtaining ALE populations with better growth in

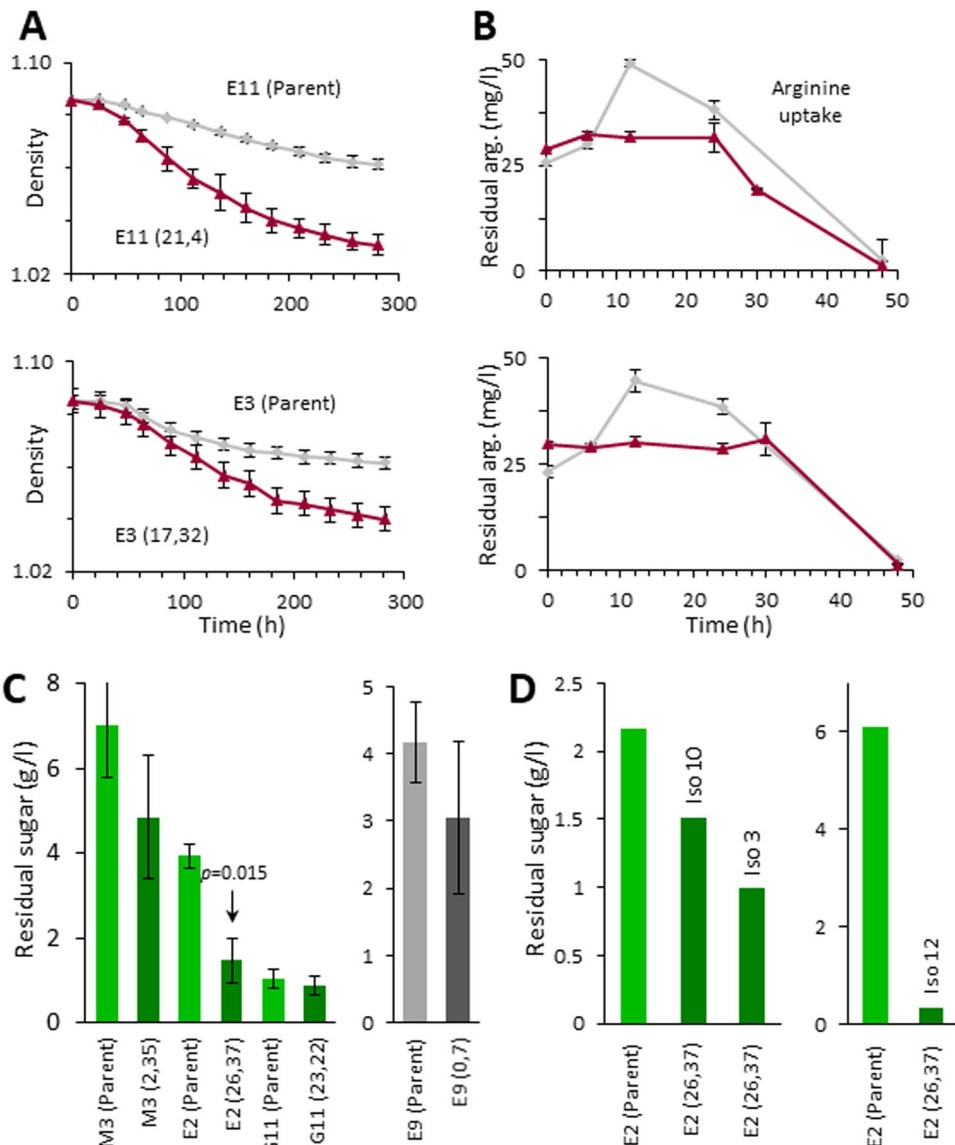

**Figure 7. ALE populations retaining adaptations in industry-like cultures.**

(A, B) We cultivated two ALE populations selected for NCR relaxation and with consistently low ammonium uptake on selection medium in larger, liquid cultures. Founder lineages are shown as references. Numbers in parenthesis: ALE population ID. (A) We tracked the fermentation (carbon dioxide production) by measuring the density of cultures. Error bars: SEM ($n = 3$ biological replicates). (B) We tracked the net arginine uptake in 80 mL cultures by measuring the residual arginine in the selection medium by high-performance liquid chromatography (HPLC). Error bars: SEM ($n = 3$ biological replicates). (C) We cultivated clones drawn from ALE populations evolved for high ethanol tolerance and three evolved for high-sugar tolerance in 5 L grape must and compared the fermentation capacity (residual sugar after 32 days) to that of their parental lineages. Error bars: SEM ($n = 3$ biological replicates). Arrow: significantly better (one-sided Student's $t$ test, $P < 0.05$). (D) We cultivated three clones drawn from ALE population E2 (26, 37) evolved for high-sugar tolerance in 80 L grape must and compared the fermentation capacity (residual sugar) to that of their parent. Residual sugar at the end (360 h) of fermentation is shown. See Appendix Fig. S5 for complete sugar consumption dynamics. Panels: Cultivations performed with distinct grape musts from different harvests (years). Source data are available online for this figure.

the ALE environment and of finding ALE populations unburdened by poor growth side effects. This agrees with population-genetic expectation. New mutations emerge stochastically, and when still young and present at only low frequencies, chance heavily affects their fates, leading to divergence between replicated ALE populations. Genetic start-point variation is expected to amplify this divergence, as the phenotypes encoded by new mutations often depend on other variants that are already present (Chiotti et al, 2014; Wang et al, 2013b), such that the latter guide evolution down

different evolutionary paths (Blount et al, 2008; Gerstein and Berman, 2020).

Arguably, the benefits of ALE parallelization depend on the specifics of the selection regime. In our current design, the relatively small population size ($< 3 \times 10^7$ cells) means that strongly beneficial mutations will manifest only relatively rarely (Sniegowski and Gerrish, 2010). In larger ALE populations, the most advantageous mutations will occur more often and this should speed up adaptation, reduce the variation in adaptation across

replicated ALE populations, and consequently also diminish the benefits of ALE parallelization. However, the strongest mutations are often associated with side effects that impair growth in non-selection niches (Chavhan et al, 2020). Moreover, they tend to be incompatible with each other (Ono et al, 2017), leaving large populations stranded on suboptimal local fitness peaks and incapable of sustaining adaptation. Our current design, ALE of many small populations, may therefore have advantages that ALE of a single, or even many, larger ALE population will struggle to match. The empirical data from our study indicates an average rate of successful improvement (> twofold improvement in growth rate) to be around $4 \times 10^{-5}$ per generation (~60 improvements among 6664 lineages, with circa 200 generations per lineage). This is remarkably low - a low throughput ALE with tens of parallel lines of a single strain would in comparable environments require multiple years for obtaining a single successful clone—attesting to the advantage presented by massively parallel experimental evolution demonstrated here.

Wine yeasts often inherit all, or the lion's share, of their genome from the Wine/European *S. cerevisiae* clade (Liti et al, 2009; Peter et al, 2018) from which our parental lineages all descend. Despite their close relatedness (mean pairwise nucleotide diversity, π: 0.00068), they nevertheless differed markedly in adaptation and side effects, in line with differences in even a single gene often altering evolution (Farkas et al, 2022). The power of highly parallelized ALE will arguably increase with greater diversity among genetic start points and inclusion of wine yeast with admixed or hybrid genomes or carrying DNA introgressed or horizontally transferred from other clades or microbes (Dequin and Casaregola, 2011; González et al, 2006), may in this context be particularly valuable. We note that an alternative ALE design, pooling strains to generate diverse starting populations (Li et al, 2019), will not exploit the initial genetic diversity to the same extent because strong early selection on the best pre-existing variants will restrict the subsequent de novo mutation-based evolutionary search (Vázquez-García et al, 2017b). Meiotic recombination offers no easy workaround (Kosheleva and Desai, 2018; McDonald et al, 2016) as domestication has impaired the sexual life cycle of most domesticated and industrial budding yeast strains (De Chiara et al, 2022).

Despite the large body of literature reporting a decelerating adaptation as populations become fitter (Couce and Tenaillon, 2015; Lukačišinová et al, 2020; MacLean et al, 2010a; Venkataram et al, 2020) because of the weaker effect of new mutations in fitter backgrounds (Chou et al, 2011; Khan et al, 2011; MacLean et al, 2010b; Wang et al, 2016), diminishing returns only explained a minor fraction of the variation in doubling-time gains among our ALE populations. Thus, our platform improved ALE outcomes across the parental lineage performance spectrum, and the ALE populations with the largest doubling-time gains often ended up being among the best performers, and superior to all parents. The moderate influence of diminishing-return adaptation is reassuring, as it demonstrates that ALE is not restricted to compensating for defects in poor-performing lineages, but can also improve the best-performing wine strains, and thus, has potentially broad versatility.

Apart from the consistent point-mutation-based inactivation of the ammonium and methylamine importer Mep2 during selection for NCR relaxation, large copy number variants (CNVs) in the form of complete or almost complete chromosome amplifications emerged as the most likely drivers of the majority of ALE adaptations, consistent with what has been observed in *S. cerevisiae* lab strains (Dunham et al, 2002; Fontanillas et al, 2010; Gresham et al, 2008; Pavelka et al, 2010; Rancati et al, 2008; Selmecki et al, 2015; Sunshine et al, 2015; Yona et al, 2012) and in drug-treated fungal pathogens (Selmecki et al, 2006; Selmecki et al, 2009). Because of their recurrence across populations, the detected CNVs are also likely to explain the emergence of side-effect syndromes, and in particular the common "good carbon use, poor nitrogen use" side effect. This syndrome may stem from parent wine strains being relatively poorly adapted to growing on synthetic wine must medium, and therefore adapting to this background environment, and in particular to the abundant (20%) sugar, during the ALE regime. ALE adaptation to concentrated sugar may reroute metabolic fluxes towards central carbon metabolism and fast fermentation, while fluxes leading away from the central carbon metabolism, to amino acid biosynthesis, are slowed. This could conceivably lead to fast growth in environments where sugar, regardless of type, at the cost of slower growth in nitrogen-limited environments. The ancient duplication of the *S. cerevisiae* genome and the concerted amplification of many glycolytic genes, is believed to explain its good use of fermentable sugars (Conant and Wolfe, 2007), and chromosome amplifications during ALE may well have a similar effect. Implicit in this conjecture is that wine yeasts are far from optimally adapted to environments rich in concentrated sugar, likely being constrained by strong historical selection for growth under nitrogen deprivation (Ibstedt et al, 2015).

Our ALE platform offered not only parallelization but also could generate improved wine strains that expressed the desired traits in larger, liquid cultures. Hence, not only does it bring evolutionary engineering into the realm of high-throughput science but may also open a new fast-track lane for optimizing microbes for industry-desired traits. And because the platform works well with microbes covering broad swaths of the tree of life, including bacteria (Alalam et al, 2020), it has the potential to be of value across many biotechnological sectors.

## Methods

### Reagents and tools table

| Reagent/resource | Reference or source | Identifier or catalog number |
|---|---|---|
| **Experimental models** | | |
| L 71B (E1) | Commercial wine yeasts | Lalvin 71B®. Isolated by INRA-Narbone (France) |
| L CLOS (E2) | Commercial wine yeasts | Lalvin CLOS®. Isolated by URV in DOQ Priorat wine region (Spain) |
| L QA23 (E3) | Commercial wine yeasts | Lalvin QA23®. Isolated by UTAD in Vinhos verdes wine region (Portugal). *S. cerevisiae bayanus* |
| LEC (E4) | Commercial wine yeasts | Lalvin EC1118®. Isolated in Champagne wine region (France). *S. cerevisiae bayanus* |
| L T73C (E5) | Commercial wine yeasts | Lalvin T73®. Isolated by IATA-CSIC in DO Alicante wine region (Spain). *S. cerevisiae bayanus* |
| U VN (E6) | Commercial wine yeasts | Uvaferm VN®. Isolated by IVICAM (Spain) in DO La Mancha wine region (Spain) |
| U BC (E7) | Commercial wine yeasts | Uvaferm BC®. Isolated by Institute Pasteur (France). *S. cerevisiae bayanus* |

| Reagent/resource | Reference or source | Identifier or catalog number |
|---|---|---|
| U BDX (E8) | Commercial wine yeasts | Uvaferm BDX®. Isolated by U. Bordeaux (France) |
| U CS2 (E9) | Commercial wine yeasts | Uvaferm CS2®. Used as the control strain throughout the phenotyping experiments. |
| U EXE (E10) | Commercial wine yeasts | Uvaferm EXENCE®. Isolated by IWB in Stellenbosch (South Africa), result of the crossing of two Sc strains. |
| U WAM (E11) | Commercial wine yeasts | Uvaferm WAM®. Isolated by U. Valladolid in DO Rueda wine region (Spain). |
| U 43 (E12) | Commercial wine yeasts | Uvaferm 43®. Isolated by Institute Inter Rhône (France). Fructofilic yeast. *S. cerevisiae bayanus* |
| U CEG (D12) | Commercial wine yeasts | Uvaferm CEG®. Isolated by Geisenheim Research Station (Germany) |
| V BMW58 (D11) | Commercial wine yeasts | Velluto BMV58®. Isolated by IATA-CSIC in DO Valencia wine region (Spain). *S. uvarum* |
| Cross Evolution (D10) | Commercial wine yeasts | Cross Evolution®. Selected by IWB in Stellenbosch (South Africa), result of the backcrossing of two Sc strains. |
| SL6 (G1) | Local cellar isolates | Natural isolate from DOQ Priorat wine region (Ferrer-Bobet winery) |
| SFB2 (G2) | Local cellar isolates | Natural isolate from DOQ Priorat wine region (Ferrer-Bobet winery) |
| SFB1 (G3) | Local cellar isolates | Natural isolate from DOQ Priorat wine region (Ferrer-Bobet winery) |
| SFB3 (G4) | Local cellar isolates | Natural isolate from DOQ Priorat wine region (Ferrer-Bobet winery) |
| SFB5 (G5) | Local cellar isolates | Natural isolate from DOQ Priorat wine region (Ferrer-Bobet winery) |
| SFB4 (G6) | Local cellar isolates | Natural isolate from DOQ Priorat wine region (Ferrer-Bobet winery) |
| SFB7 (G7) | Local cellar isolates | Natural isolate from DOQ Priorat wine region (Ferrer-Bobet winery) |
| SFB6 (G8) | Local cellar isolates | Natural isolate from DOQ Priorat wine region (Ferrer-Bobet winery) |
| SFB10 (G9) | Local cellar isolates | Natural isolate from DOQ Priorat wine region (Ferrer-Bobet winery) |
| SFB9 (G10) | Local cellar isolates | Natural isolate from DOQ Priorat wine region (Ferrer-Bobet winery) |
| SFB8 (G11) | Local cellar isolates | Natural isolate from DOQ Priorat wine region (Ferrer-Bobet winery) |
| SL4 (G12) | Local cellar isolates | Natural isolate from DOQ Priorat wine region (Ferrer-Bobet winery) |
| SL3 (F12) | Local cellar isolates | Natural isolate from DOQ Priorat wine region (Ferrer-Bobet winery) |
| M2 (M2) | Local cellar isolates | Natural isolate from DOQ Priorat wine region (Mas Perinet winery) |
| M3 (M3) | Local cellar isolates | Natural isolate from DOQ Priorat wine region (Mas Perinet winery) |
| M4 (M4) | Local cellar isolates | Natural isolate from DOQ Priorat wine region (Mas Perinet winery) |
| M5 (M5) | Local cellar isolates | Natural isolate from DOQ Priorat wine region (Mas Perinet winery) |
| M6 (M6) | Local cellar isolates | Natural isolate from DOQ Priorat wine region (Mas Perinet winery) |
| M7 (M7) | Local cellar isolates | Natural isolate from DOQ Priorat wine region (Mas Perinet winery) |
| M9 (M9) | Local cellar isolates | Natural isolate from DOQ Priorat wine region (Mas Perinet winery) |
| M10 (M10) | Local cellar isolates | Natural isolate from DOQ Priorat wine region (Mas Perinet winery) |
| M11 (M11) | Local cellar isolates | Natural isolate from DOQ Priorat wine region (Mas Perinet winery) |
| M12 (M12) | Local cellar isolates | Natural isolate from DOQ Priorat wine region (Mas Perinet winery) |
| M13 (M13) | Local cellar isolates | Natural isolate from DOQ Priorat wine region (Mas Perinet winery) |
| T3 (T3) | Local cellar isolates | Natural isolate from DO Terra Alta wine region |

| Reagent/resource | Reference or source | Identifier or catalog number |
|---|---|---|
| T4 (T4) | Local cellar isolates | Natural isolate from DO Terra Alta wine region |
| T5 (T5) | Local cellar isolates | Natural isolate from DO Terra Alta wine region |
| T6 (T6) | Local cellar isolates | Natural isolate from DO Terra Alta wine region |
| T7 (T7) | Local cellar isolates | Natural isolate from DO Terra Alta wine region |
| T8 (T8) | Local cellar isolates | Natural isolate from DO Terra Alta wine region |
| T14 (T14) | Local cellar isolates | Natural isolate from DO Terra Alta wine region |
| T15 (T15) | Local cellar isolates | Natural isolate from DO Terra Alta wine region |
| T19 (T19) | Local cellar isolates | Natural isolate from DO Terra Alta wine region |
| **Recombinant DNA** | | |
| **Antibodies** | | |
| **Oligonucleotides and sequence-based reagents** | | |
| PCR primers: delta 12 5'-TCAACAATGGAATCCCAAC-3' | Eurofins | |
| PCR primers: delta 21 5'CATCTTAACACCGTATATGA-3' | Eurofins | |
| **Chemicals, enzymes, and other reagents** | | |
| Gelrite (gellan gum) | Sigma-Aldrich | G1910 |
| 2-Deoxy-D-glucose | Sigma-Aldrich | D6134 |
| Diamide | Sigma-Aldrich | D3648 |
| Methylamine hydrochloride | Sigma-Aldrich | M0505 |
| **Software** | | |
| *PRECOG* | http://precog.lundberg.gu.se/Pages/Content/GettingStarted | |
| *Fastqc v. 0.11.4* (https://www.bioinformatics.babraham.ac.uk/projects/fastqc) | https://www.bioinformatics.babraham.ac.uk/projects/fastqc | |
| *cutadapt v. 1.10* | https://pypi.org/project/cutadapt/ | |
| *Picard Tools v. 1.129* | https://broadinstitute.github.io/picard/ | |
| *Scan-o-matic, version 1.5.7* | https://github.com/Scan-o-Matic/scanomatic | |
| *Matlab (R2017b v. 9.3.0, R2019b, v. 9.7.0)* | https://www.mathworks.com/products/matlab.html | |
| R v. 4.0.3 | https://www.r-project.org | |
| Python (v. 3.6, v. 3.6.13) | https://www.python.org/ | |
| fastSTRUCTURE v. 1.0 | https://rajanil.github.io/fastStructure/ | |
| GATK4 Mutect2 v. 4.1.0.0 | https://gatk.broadinstitute.org/hc/en-us | |
| **Other** | | |
| NEBNext DNA Ultra2 Library Preparation Kit | New England Biolabs, USA | |

| Reagent/resource | Reference or source | Identifier or catalog number |
|---|---|---|
| Epson Perfection V800 PHOTO scanners | Epson Corporation, UK | |
| Kodak Professional Q-60 Color Input Target | Kodak Company, USA | |
| 2100 BioAnalyzer | Agilent Technologies | |
| HiSeq 2000, HiSeq2500 | Illumina | |
| Singer RoToR HDA robot | Singer Instruments, UK | |
| SBS-format PlusPlates | Singer Instruments, UK | |
| Oxygen-permeable film (Breathe-Easy). | Sigma-Aldrich | Z380059 |
| Glutathione assay kit | Sigma-Aldrich | CS0260 |
| Bioscreen C | Growth Curves Ltd, Finland | |
| Epicentre MasterPure Yeast DNA Purification Kit | | |

## Strains and growth medium

We obtained 15 commercial wine yeasts marketed by Lallemand Inc. (Canada) and 33 noncommercial wine yeasts. We isolated the latter from grapes or vineyard soil in the DOQ Priorat wine-making region in Catalonia and identified them as *S. cerevisiae* using restriction fragment length polymorphisms (Padilla et al, 2016). Strains are listed in Appendix Table S1. Strains were stored long-term at -80 C in 20% (v/v) glycerol. Experiments in Figs. 2–6 and EV1–4; Appendix Figs. S1–S3 were performed in synthetic grape must medium(Beltran et al, 2004). Vitamins (100×; pH adjusted to 3.3 with NaOH), amino acids (10×; pH adjusted to 3.3 with NaOH, buffered with 2% (w/v) $Na_2CO_3$), oligo-elements (1000×) and anaerobiosis factors (1000×; ergosterol, oleic acid, Tween 80 and ethanol) were stored as separate, sterile filtered stock solutions at 4 °C (amino acids <2 weeks; vitamins at −20 °C). To prepare the synthetic grape must, glucose (100 g/L), fructose (100 g/L), citric acid (5 g/L), malic acid (0.5 g/L), tartaric acid (3 g/L), $KH_2PO_4$ (0.75 g/L), $K_2SO_4$ (0.5 g/L), $MgSO_4.7H_2O$ (0.25 g/L), $CaCl_2.2H_2O$ (0.155 g/L), NaCl (0.2 g/L), and NH4Cl (0.153 g/L) were dissolved in $H_2O$, autoclaved, and pH were set to 3.3 with NaOH. For solid medium experiments, a separate solution of the gelifying agent gelrite (gellan gum), which have better retention of water, less phenolic contamination and better light transmission properties than the classical agent agar (Huang et al, 1995; Jaeger et al, 2015; Lin and Casida, 1984), and $CaCl_2.2H_2O$ (initiates gelification) were then prepared, adjusted to pH=3.2 (NaOH), autoclaved, and added to the medium (final concentrations: gelrite=8 g/L, $CaCl_2.2H_2O$ = 1.155 g/L). The volume was set to 1 L (w. sterile $H_2O$). For solid medium, the medium was stirred on a heater (at 50 °C) and 50 mL was poured into the same lower left corner of each Plus plate (Singer, UK). Plates were dried overnight and used, with no additional storage. For particular experiments, variations to the basic synthetic grape must were made, as described in Appendix Tables S2 and S3. Scale-up experiments (5 L and 80 L) shown in Fig. EV5 and Appendix Figs. S4 and S5 were performed in White Grenache (GR) grape must harvested just before experiment start in D. O. (Denomination of Origin) Terra Alta (Spain).

## Adaptive laboratory evolution (ALE)

Each of the 48 parent strains to be ALE evolved was stored as two separate populations, both of which had been clonally expanded from the same single cell, in 20% (v/v) glycerol in 96-well format. The 96 frozen stocks were thawed, re-pinned 12× onto solid, synthetic grape must medium using long-pin pads (Singer) to generate 1152 colonies in a 1536 array and cultivated for 72 h, all as shown in Appendix Fig. S1. This pre-culture was repeated once to further standardize the physiological states of populations. We maintained one in every four positions empty to continuously survey cross-contamination and to allow inclusion of fixed controls in the growth measurement stage. Standardized pre-cultures were re-pinned (1536 short pin transfer) onto eight ALE selection environments to generate in total 9216 ALE yeast populations. The eight selection media were synthetic grape must with: (i) 20% (w/v) fructose as the sole sugar, together with 2 g/L 2-deoxyglucose (fructose utilization), (ii) 10% of the regular amino acid concentration, i.e., 10 mg N/L (nitrogen starvation), (iii) glycine, glutamine and cysteine as the sole nitrogen sources, together with 1.5 mM diamide (glutathione production), (iv) arginine and proline as the sole nitrogen sources together with 1% (w/v) methylamine (Nitrogen Catabolite Repression (NCR) relaxation), (v) 35% (w/v) sugar with equal proportion of fructose and glucose (high-sugar tolerance), (vi) 1.3% (v/v) 1-butanol (ethanol tolerance), (vii) valine, iso-leucine, and phenylalanine as the sole nitrogen sources (aroma production), and (viii) 1% of the regular vitamin concentration (vitamin starvation). The ALE environments are further described in Appendix Table S2. Synthetic grape must was used as pre-culture for the selection media (i), (v), and (vi), while synthetic grape must with low nitrogen (N) content (30 mg N/L) was the pre-culture for the rest of the selection media. We passed populations through a batch-to-batch selection regime of 30 consecutive growth cycles. In each cycle, we clonally expanded populations for 72 h from around $n = 10^5$ cells. We subsampled the mostly stationary phase populations using 1536 short pin transfers, and deposited samples ($n = 10^5$ cells) on fresh selection medium. Each growth cycle corresponded to a mean of 6.6 population doublings for a total of ~200 doublings (corresponding to cell generations, assuming no cell death). We observed no invasions of empty colony positions during any stage of the evolution, or cultivation after storage, process. This is consistent with what has been observed in earlier papers relying on the same protocol (Persson et al, 2022; Stenberg et al, 2022) and suggest that cross-contamination in association with the ALE protocol is extremely rare. There is a nonzero risk for contamination in the manual handling and liquid cultivation steps required for follow-up experiments. To estimate this risk, we manually revived and cultivated 65 fast-adapting endpoint populations and their parents in liquid medium, extracted their DNA and performed a restriction fragment length polymorphism (RFLP) of interdelta elements. This showed that a substantial majority ($n = 56$) of adapted lineages retained the parental RFLP pattern, whereas an unequivocal call could not be made for the remaining nine. Thus, we cannot exclude that contamination affects a small minority of our follow-up experiments.

The ALE procedure was standardized to minimize bias as follows. Solid media was cast in polystyrene plates from a single batch (Singer Instruments, SBS-format PlusPlates). Each plate was

cast on top of the same perfectly level surface with precisely 50 mL of synthetic grape must medium at 50 °C. Medium was poured in the same lower left corner on all plates. All plate preparations were performed in the same environmentally controlled space (room temperature 23 °C). Pinning transfers were performed using 1536 short pin pads from the same production batch (Singer Instruments, UK), and a Singer RoToR HDA robot (Singer Instruments, UK). All transfers were performed using a single robot, standing in the same environmentally controlled space throughout the experimental series and a single production batch of Plus plates and pin pads. The robotic workspace was sterilized by prolonged UV exposure before each use and routinely cleaned with ethanol. All evolution plates were kept in the same temperature-controlled (30.0 C) thermostatic cabinet. We stored a frozen fossil record of populations in glycerol in 96-well plates, by pinning from agar to liquid medium using the Singer RoToR HDA robot. After 3 days of incubation at 30 °C, we added glycerol to a final concentration of 20% (v/v) and stored samples at −80 °C.

## Counting cells in growing parents and ALE endpoint populations

To monitor population size expansion of parent ($t = 0$ growth cycles) and endpoint ($t = 30$ growth cycles) populations, we used a high-resolution microbial growth phenotyping platform, Scan-o-matic, version 1.5.7 (Zackrisson et al, 2016). First, we measured the growth of parent populations in many environments at high replication ($n = 24$) by thawing of frozen stocks, pre-cultivation (2×), subsampling and deposition on fresh plates, as described above. Second, we thawed frozen stocks of parent and endpoint populations, pre-cultivated these (2×) in parallel, subsampled pre-cultures deposited subsamples on each of 18 different environmental plates and measured their doubling time. Experimental populations in the two set-ups were handled and analyzed identically. Using a custom-made RoToR pinning program, we deposited 384 control samples, subsampled from a 384 pre-culture array of genetically identical control colonies (strain E9), in the 384 interleaved empty positions on each target plate. We recorded population size at 20 min intervals for 3 days using high-quality desktop scanners (Epson Perfection $V_{800}$ PHOTO scanners, Epson Corporation, UK) connected via USB to a standard desktop computer. We performed transmissive scanning at 600 dpi using 8-bit gray-scale. Plates were fixed in place by custom-made acrylic glass fixtures. Pixel intensities were normalized and standardized across instruments using transmissive scale calibration targets (Kodak Professional Q-60 Color Input Target, Kodak Company, USA) which were fixed to each fixture. Pixel intensities were estimated and summed across each colony, the local background was subtracted, and pixel intensities due to cells were converted into cell counts by calibration to independent cell number estimates obtained using spectrometer and flow cytometry. Raw measurements of population size were smoothed, and the steepest slope in each growth curve was converted into a population size doubling time. Noisy growth curves were flagged, visually inspected for artifacts and 0.3% of doubling-time estimates were rejected as potentially incorrect. Doubling times were $\log_2$ transformed and normalized to those estimated for each position using the 384 spatial controls on each plate, thereby producing $\log_2$

doubling-time ratios that account for systematic variations in doubling times across and between plates. We estimated the adaptation for each ALE population as the difference in normalized doubling time between its evolution endpoint ($t = 30$ growth cycles) and start point ($t = 0$ growth cycles). We estimated how much of the total variance in adaptation, across all ALE populations, that can be explained by variance between selection environments, between strains (genotypes), and between replicate ALE of individual strains. We projected the data into the selection dimension, the strain dimension, and the replicate dimensions respectively, and calculated their corresponding variances. We extracted the technical variance as the residual variance after other variances had been subtracted from the total variance. We expressed the estimated variances as the fraction of the total variance they explain, by dividing with the total variance. We calculated t-distributed stochastic neighbor embedding (t-SNE) of strains based on similarity in side effect using Matlab and separated clusters using k-means, with the k specified by counting the number of clusters by hand.

## Sequencing and sequence analysis

Total genomic DNA was extracted from parental lineages grown in YPD using phenol-chloroform-based extraction and from evolved populations grown in YPD using Epicentre MasterPure Yeast DNA Purification Kit. The DNA quality was evaluated with electrophoresis in a 1% (w/v) agarose gel and DNA concentrations were evaluated using Qubit (Thermo Fisher Scientific, USA). An equal amount of DNA from each sample was used for library preparation with the NEBNext DNA Ultra2 Library Preparation Kit (New England Biolabs). The library preparation was performed on an automated liquid handling system (Hamilton Robotics) and the quality of the library was tested on a 2100 BioAnalyzer (Agilent Technologies). Paired-end Illumina short read sequencing was performed at the Genomics Core Facility (EMBL Heidelberg) on HiSeq 2000 and HiSeq2500 platforms (Illumina, San Diego, USA) for 150 bp (average insert size: 245 bp) and 250 bp (average insert size: 616 bp) reads, respectively. The data is deposited in the European Nucleotide Archive (ENA) at EMBL-EBI under accession number PRJEB41108 (https://www.ebi.ac.uk/ena/browser/view/PRJEB41108).

The quality of the sequencing reads was controlled using Fastqc v. 0.11.4 (https://www.bioinformatics.babraham.ac.uk/projects/fastqc/; Andrews, 2010). The reads were trimmed by removing adapters and filtering low-quality reads using cutadapt v. 1.10 (Martin, 2011; https://pypi.org/project/cutadapt/). The trimmed reads were aligned to the S. cerevisiae EC1118 wine yeast regenome assembly (Novo et al, 2009) with the Burrows-Wheeler Aligner v. 0.7.12 (Li, 2013; Li and Durbin, 2009) using default parameters. Picard Tools v. 1.129 (https://broadinstitute.github.io/picard/) were used to process (read groups addition, sorting, reordering, and indexing) the alignments and mark duplicate reads.

Single-nucleotide polymorphism (SNP), and insertion–deletion (indel) variant calling of parent samples (excluding strain D11 with low-quality sequencing sample) was performed with GATK4 v. 4.1.0.0 HaplotypeCaller in GVCF model using the S. cerevisiae EC1118 (Novo et al, 2009) as the reference with DISCOVERY genotyping mode, ploidy 2, and minimum base quality score 20. The individual GVCF files were then combined using

*CombineGVCFs*, and jointly genotyped using *GenotypeGVCFs* with ploidy 2 and standard minimum confidence of calling 20. Using *SelectVariants* and *VariantFiltration* tools the called SNPs and indels were filtered separately. SNPs were filtered with QD < 2.0, FS > 60.0, MQ < 40.0, MQRankSum < −12.5, ReadPosRankSum < −8.0, GQ < 30, DP < 5. Indels were filtered with QD < 2.0, FS > 200.0, MQ < 40.0, ReadPosRankSum < -20.0, GQ < 30, DP < 5. Then, the SNP sites were further filtered using vcftools v. 0.1.14 for sites which miss more than 50% of the data (max-missing, 0.5), for other than biallelic sites (max-alleles, 2; min-alleles, 2), for sites with minor allele frequency less than 0.05 (maf, 0.05), and for sites not in Hardy–Weinberg equilibrium (HWE, 0.00001). To estimate average nucleotide diversity among the parental lineages vcftools v. 0.1.14 was used. A neighbor-joining tree of the parental lineages was created from the filtered set of 42599 segregating sites using R v. 4.0.3 (R Core Team (2020). R: A language and environment for statistical computing. R Foundation for Statistical Computing) and packages ape v. 5.4.1 (https://github.com/emmanuelparadis/ape) and SNPrelate v. 1.24.0 (http://github.com/zhengxwen/SNPRelate). First, individual dissimilarities (derived from relative coancestry) were estimated for each pair of individuals with the *snpgdsDiss* function. The individual dissimilarities were then used as distances for the *bionj* function. Model-based Bayesian algorithm fastSTRUCTURE v. 1.0(Raj et al, 2014) was used to detect and quantify admixture in the 47 parental lineage genomes (excluding strain D11 with low-quality sequencing sample). fastSTRUCTURE was run on the filtered set of 42,599 segregating sites, varying the number of parent populations ($K$) between 1 and 10 using the simple prior implemented in fastSTRUCTURE. $K = 4$ was found to be optimal, i.e., scoring the highest marginal likelihood.

Single-nucleotide variant (SNV), and small insertions-deletions (indel) in ALE populations were called using GATK4 Mutect2 v. 4.1.0.0 (Van der Auwera et al, 2013), default parameters and a panel-of-normals approach. We created the high-coverage sequence panel using 47 parental lineages (excluding D11) and the *CreateSomaticPanelOfNormals* tool. We called variants in each ALE endpoint relative to the panel using *FilterMutectCalls* and default parameters, including a tumor-lod of 5.3. Copy number variants in ALE populations (CNV) were called analysis using ATK4 v. 4.1.0.0 tools. Read for 1000 bp intervals were counted with *CollectReadCounts and* read counts were denoised and matched to that of its corresponding parental lineage sample using *DenoiseReadCounts*. The allelic counts were collected using *CollectAllelicCounts* and combined with the binned read count ratios for modeling the CNV segments using *ModelSegments* with number-of-change-points-penalty-factor of one (1) or ten (10) for HiSeq 2000 and HiSeq2500 samples respectively. CNVs were called using *CallCopyRatioSegments*. Called CNV segment copy ratios were further re-centered to the corresponding sample medians. The sample H2, with a high noise copy ratio comparing to the corresponding parental lineage and whose contigs were smaller than 10 kb was excluded. Loss-of-heterozygosity in ALE populations was called as *baf* zero/one segments across sites for which the corresponding parental lineage was called as heterozygotic (baf~0.5). The degree of heterozygosity in parental strains was calculated as a ratio of observed heterozygotic snps to observed homozygotic snps using vcftools v. 0.1.16 --*het* (Danecek et al, 2011).

## Fermentation validation of ALE endpoints

Selected ALE endpoint populations were validated at lab scale by fermenting them and their respective parental lineages in a synthetic grape must (pH 3.3) as described in Beltran et al (Beltran et al, 2004), with the nitrogen content varying between fermentations. The control synthetic grape must contain 200 g/L of sugar (100 g/L glucose and 100 g/L fructose) and 300 mg of yeast assimilable nitrogen/L (120 mg/L inorganic and 180 mg/L organic nitrogen).

All fermentations were performed in biological triplicates at 22 °C with agitation (120 rpm) in laboratory-scale fermenters: 250-mL flasks containing 200 mL of medium and capped with closures that enabled carbon dioxide to escape and samples to be removed. Fermentations were performed in semi-anaerobic conditions, with small amounts of oxygen entering cultivation flasks during sampling only. The initial yeast inoculum consisted of $2 \times 10^6$ cells/mL taken from YPD stationary phase cultures (48 h). Sugar consumption was monitored throughout the fermentation process by measuring decrease in fermentation medium density using a densitometer (Densito 30PX, Mettler-Toledo, Switzerland). In the later stages of the fermentation, when the sugar contribution to medium density is limited, sugar consumption (glucose and fructose) was assayed using enzymatic kits (Roche Applied Science, Germany). Fermentation was considered to be completed when residual sugars were below 2 g/L. Yeast cell counts were determined by measuring optical density at 600 nm. Yeasts cells were harvested at different time points for measurement of the nitrogen, sugar or total glutathione content. Cell pellets were transferred to Eppendorf tubes, frozen immediately in liquid nitrogen and kept at −80 °C until they were analyzed. The supernatant was stored at −20 °C for later analysis of their remaining nitrogen content.

## Low-volume liquid cultivations

We tested the ethanol tolerance of selected strains evolved for better ethanol tolerance in low-volume liquid cultivations using Bioscreen micro-cultivation stations (Bioscreen Inc.) and followed the growth of cultures at 20 min intervals for 72 h. Honeycomb microplates were loaded with 120 μL synthetic grape must containing either 8% (v/v) ethanol or 1.3% (v/v) n-butanol. To remove any spatial bias from differences in growth conditions between well positions, e.g., in the form of e.g., ethanol evaporation, plates were run without lids but covered with an oxygen-permeable film (Breathe-Easy; Sigma-Aldrich). Initial $OD_{600} \approx 0.2$ was set for inoculation. Data was extracted from growth curves using PRECOG (Fernandez-Ricaud et al, 2016).

## Semi-industrial validations of ALE endpoints

Semi-industrial validations were performed using clonal colonies expanded from single cells isolated from selected ALE endpoint populations and their corresponding parental lineages, as listed in Appendix Table S4. To select clones for 80 L scale-up experiments, we subsampled the E2 (26, 37) ALE population, streaked for single cells on synthetic grape must, and selected 11 colonies clonally expanded from these single cells. We cultivated subsamples of these 11 colonies on synthetic grape must, while measuring their cell doubling time using the high-throughput solid medium growth

platform, as described above. Semi-industrial fermentations were conducted in 100 L stainless steel tanks filled with 80 L of White Grenache (GR) grape must from D. O. Terra Alta (Spain), as well as in 5 L fermenters, filled with 4 L of the same grape must. The musts had a density around 1100 g/L, pH = 3.31, and an initial yeast assimilable nitrogen content of 163.8 mg/L. 80 L fermentations (n = 1) and 5 L fermentations (n = 3). From each tank and fermenter, daily samples were taken to monitor sugar concentration by measuring must density using an electronic densitometer (Mettler-Toledo S.A.E., Barcelona, Spain), as well as yeast growth (by counting of colony forming units). At the middle and endpoint of the fermentation, *S. cerevisiae* isolates were typified by the analysis of interdelta regions, as described by Legras and Karst (2003) (Legras and Karst, 2003) using primers Δ12 and Δ21. The final wines were stabilized for 30 days at 4 °C, 30 ppm of $SO_2$ was added as potassium metabisulfite, and the product was bottled and stored for 2 months until the sensory evaluation took place.

## Nitrogen content analysis

For experiments in Fig. EV5B, selected ALE endpoint populations and their corresponding parents were cultivated as 80 mL cultures. Free amino acids and ammonia were analyzed using DEEMM derivatizations as in Gomez-Alonso et al (Gómez-Alonso et al, 2007) and with the Agilent 1100 Series high-performance liquid chromatography (HPLC) (Agilent Technologies, Germany). Separation was performed in an ACE HPLC column (C18-HL) particle size 5 μm (250 mm × 4.6 mm) thermostatized at 20 °C. Several dilutions of each sample were analyzed and averaged. The concentration of each compound was calculated using an internal standard, Agilent ChemStation (Agilent Technologies) and expressed as mg N/L.

## Glutathione measurement

For experiments in Fig. EV5A, total glutathione levels (intracellularly and extracellularly) were established in the stationary phase, in 18 ALE endpoint populations and their parental lineages (Appendix Table S5). Cells were pre-cultivated on 50 mL of glutathione evolution medium (see above) at 28 °C with orbital shaking (120 rpm) and after 48 h growing, cells were inoculated in 45 mL of the same medium at initial $OD_{600} = 0.2$ (n = 2). Yeast growth and media density were evaluated continuously by spectrometry and electronic densitometry and glutathione was measured after one week, with all cultures being well into the stationary phase.

For glutathione extraction, the method described by Vázquez et al, 2017 was used (Vázquez et al, 2017). Briefly, a cell density of $1 \times 10^8$ cells was centrifuged at 10,000 rpm to obtain a packed cell pellet and the supernatant. Pellets were weighed and three volumes of 5% (v/v) 5-sulfosalicylic acid (SSA) were added and vortexed. The cell suspensions were then frozen and thawed three times (using liquid nitrogen to freeze and a 37 °C bath to thaw), incubated for 5 min at 4 °C and centrifuged at 800 × *g* for 10 min at 4 °C. The measurement of total glutathione (tGSH) was determined (in pellets and supernatants) using the kinetic glutathione assay kit (Sigma-Aldrich, Spain) in which catalytic amounts (nmoles) of reduced glutathione (GSH) cause a continuous reduction of 5,5′-diothiobis-2-nitrobenzoic acid to 5-thio-2notrobenzoic acid (TNB), and the oxidized glutathione (GSSG) formed is recycled by glutathione reductase and NADPH. The final TNB formed (equivalent to tGSH) was measured spectrophotometrically at 412 nm. Other pellets ($1 \times 10^8$ cells) were previously dried at 28 °C for 48 h and weighed. Thus, the results are expressed as nmoles tGSH/mg dry weight.

## Data availability

Data presented in figures are available as Source Data files (Figs. 1–4 and 6) and in Dataset EV1, Table EV1 (Fig. 5). Data presented in Expanded View and Appendix figures are available as EV source data files. All datasets produced in this study can be downloaded from the following databases: Sequence data: European Nucleotide Archive (ENA) at EMBL-EBI, PRJEB41108, https://www.ebi.ac.uk/ena/browser/view/PRJEB41108. All other data: Mendeley, https://data.mendeley.com/datasets/685y557cth/1. Code: All analyses were performed using published code, as described in "Methods".

The source data of this paper are collected in the following database record: biostudies:S-SCDT-10_1038-S44320-024-00059-0.

## Peer review information

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

## Acknowledgements

This work was sponsored by the ERASysAPP project WINESYS (the German Ministry of Education and Research grant no. 031A605; the Research Council of Norway (Norges Forskningsråd) grant no. 245160, the Swedish Research Council grant no. 325-2014-6547) and by the Ministry of Science, Innovation and Universities, Spain (España, Ministerio de Ciencia e Innovaciòn (MCIN)) (Project CoolWine, PCI2018-092962), under the call ERANET ERA COBIOTECH. PJ acknowledges funding from the Academy of Finland, decision numbers 310514, 314125, and 329930. KRP received funding from the European Research Council (ERC) under the European Union's Horizon 2020 research and innovation programme (Grant Agreement No. 866028). We acknowledge the support of the Genomics core facilities at the European Molecular Biology Laboratory (Heidelberg, Germany).

## Author contributions

**Payam Ghiaci**: Formal analysis; Investigation; Visualization; Methodology; Writing—original draft; Writing—review and editing. **Paula Jouhten**: Formal analysis; Investigation; Visualization; Methodology; Writing—review and editing. **Nikolay Martyushenko**: Formal analysis; Visualization; Methodology. **Helena Roca-Mesa**: Investigation. **Jennifer Vazquez**: Investigation. **Dimitrios Konstantinidis**: Investigation. **Simon Stenberg**: Investigation. **Sergej Andrejev**: Investigation. **Kristina Grkovska**: Investigation. **Albert Mas**: Conceptualization; Supervision; Writing—review and editing. **Gemma Beltran**: Conceptualization; Supervision; Writing—review and editing. **Eivind Almaas**: Conceptualization; Supervision; Funding acquisition; Writing—review and editing. **Kiran R Patil**: Conceptualization; Formal analysis; Supervision; Funding acquisition; Project administration; Writing—review and editing. **Jonas Warringer**: Conceptualization; Supervision; Methodology; Writing—original draft; Writing—review and editing.

Source data underlying figure panels in this paper may have individual authorship assigned. Where available, figure panel/source data authorship is listed in the following database record: biostudies:S-SCDT-10_1038-S44320-024-00059-0.

## Funding

## Disclosure and competing interests statement

# Expanded View Figures

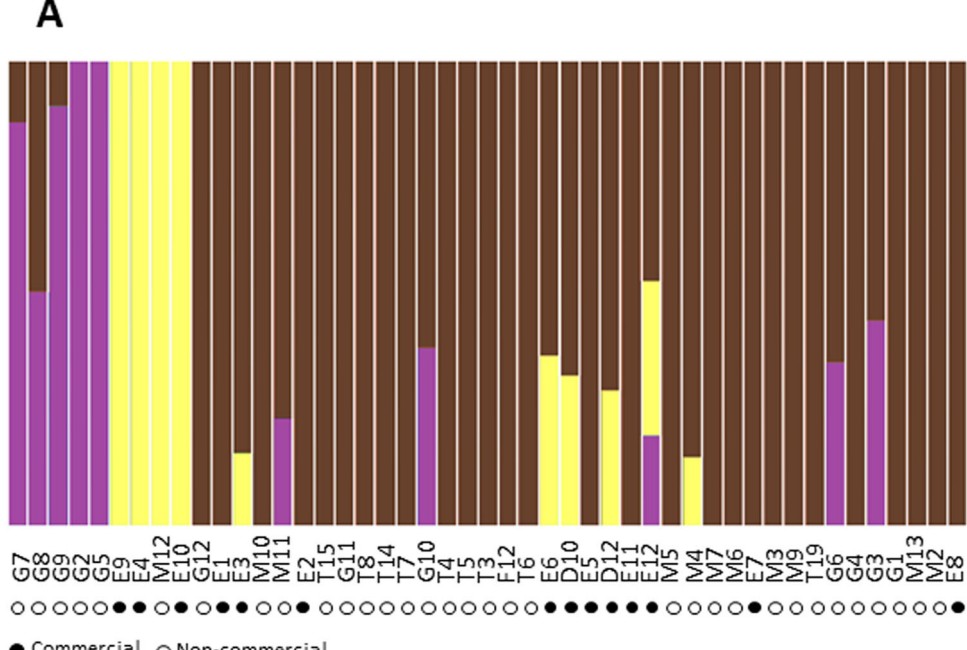

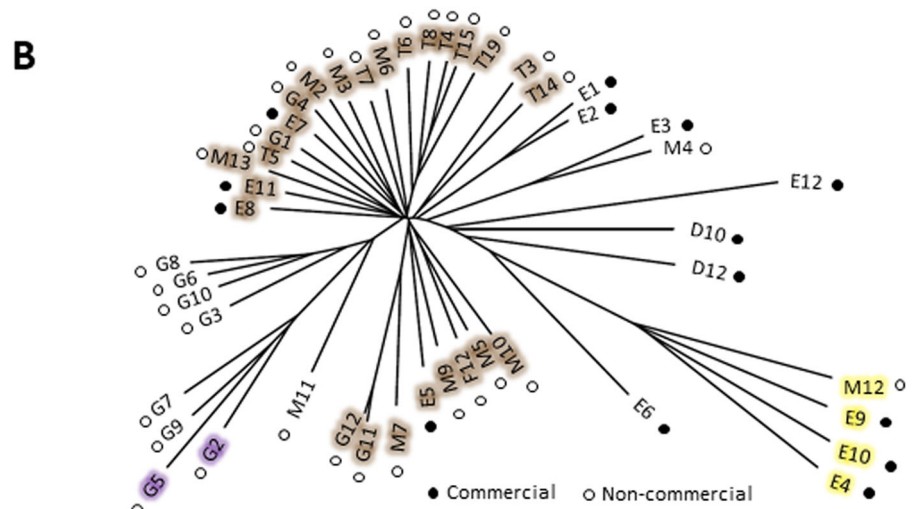

**Figure EV1. Population structure of parent wine strains, before ALE.**

(A) We sequenced the genome of 47 commercial (●) and noncommercial (○) euploid diploids wine yeast lineages (Appendix Table S1) using Illumina and called 42,599 SNPs against the Lalvin EC1118® wine strain reference genome. We varied the number of populations (K) between 1 and 10 and show the population structure and admixture for K = 4 (color). (B) Phylogeny (neighbor-joining tree) of 47 wine yeast lineages. Text color = non-admixed genetic groups identified by the admixture mapping in (A). Admixed strains are left uncolored. Bar = SNP distance.

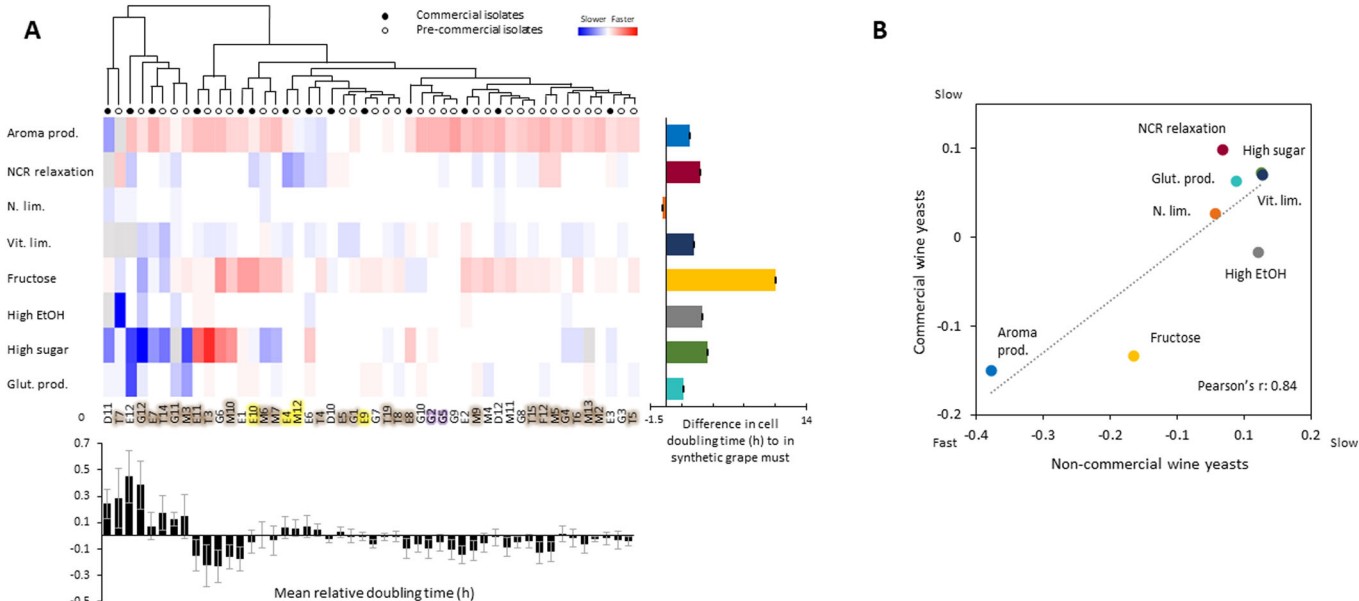

**Figure EV2. Strength of ALE selection.**

Cell doubling times of wine yeasts in selection environments. (**A**) *Central panel (heatmap):* Cell doubling times (mean: $n = 24$ replicate ALE populations) of each wine yeast normalized to the fixed control (color). No color: fixed control. Wine yeast names are colored based on population (see Fig. EV1). *Upper panel:* Hierarchical clustering of wine yeast based on similarity in cell doubling times, using Pearson's $r$ and strain averages (for groups). *Left panel:* Variance across wine yeasts ($n = 48$ strains) in mean cell doubling time ($h$). *Right panel:* Strength of selection, estimated as mean ($n = 48$ strains) difference ($h$) in cell doubling time in an environment, as compared to in synthetic grape must (SGM). Error bars = SEM. *Bottom panel:* Some wine yeasts are general slow growers, reflecting limited adaptation to synthetic grape must. Mean of normalized cell doubling times for each wine yeast, across all selection environments. Error bars = SEM ($n = 8$ environments). (**B**) Commercial ($n = 15$ strains) and noncommercial ($n = 33$ strains) wine yeast grow equally well in all selection environments. Mean cell doubling times normalized to the fixed control are shown. Broken line = linear regression. Source data are available online for this figure.

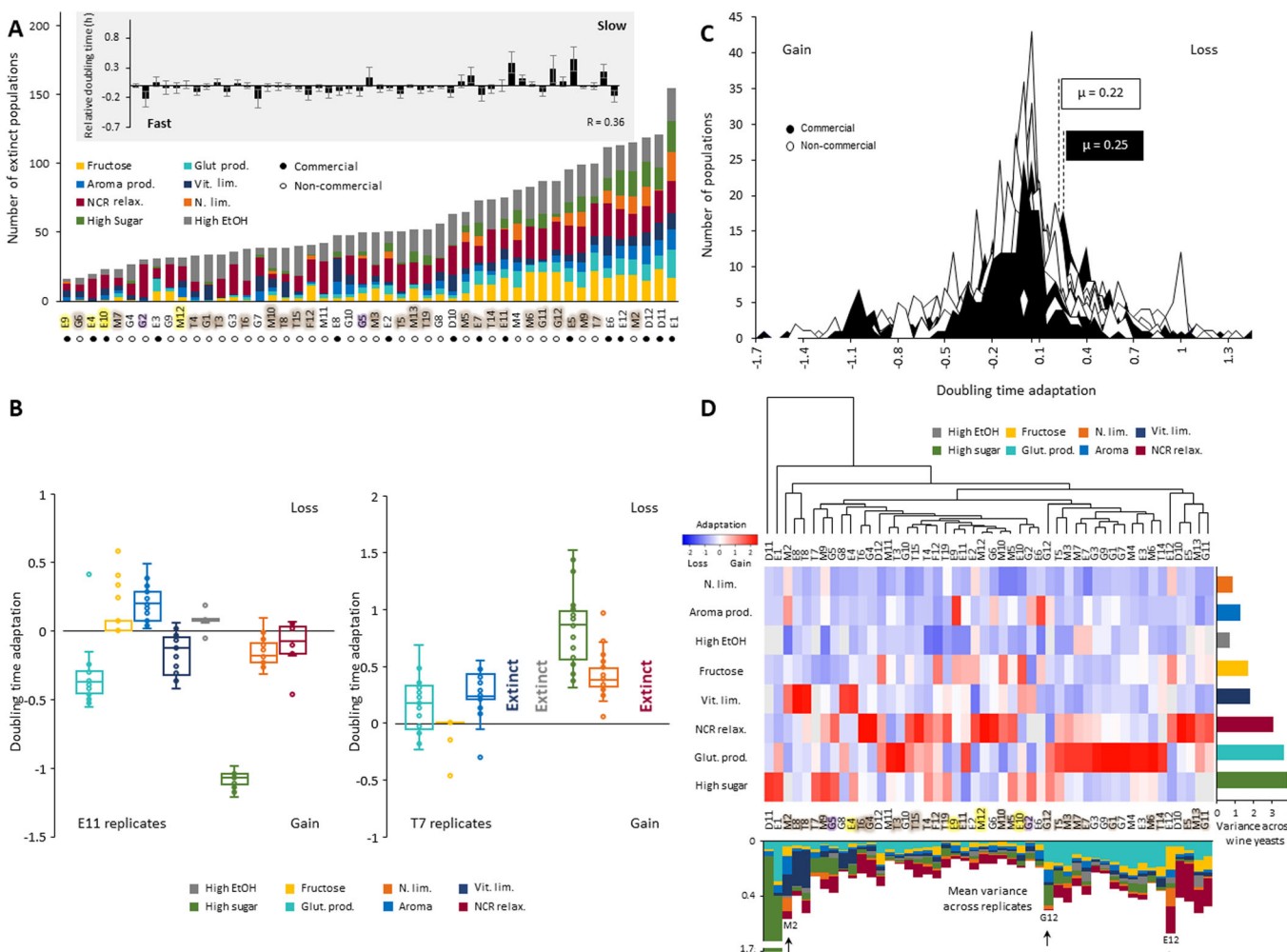

**Figure EV3. Adaptation of ALE wine yeasts.**

(A) Extinct populations: stacked bar plot of number of extinct populations, in each environment. Inset: mean of ($\log_2$, normalized) cell doubling time for each wine yeast ($n = 24$ replicate ALE populations, extinct populations excluded, in each environment) across all environments ($n = 8$ environments). Error bars: SEM ($n = 8$). (B) Adaptation ($n = 24$ replicate ALE populations, extinct populations excluded) of wine yeasts E11 (left panel) and T7 (right panel) in selection environments (color). Box: interquartile range, line: median: whiskers 1.5x interquartile range, outliers: populations outside interquartile range. (C) Histogram of adaptation for all commercial (black, $n = 2880$ ALE populations) and noncommercial (white, $n = 6336$ ALE populations) ALE populations, totaled across all selection regimes and wine yeast lineages. Extinct populations are excluded. (D) Adaptability has a genotype-by-environment component. *Central panel (heatmap):* Adaptation (mean: $n = 24$ replicate ALE populations) of each wine yeast. No color: no adaptation. Wine yeast names are colored based on population (see Fig. EV1). *Upper panel:* Hierarchical clustering of wine yeast (Pearson's $r$, averages used to cluster groups) based on similarity in adaptation across eight selection regimes. *Right panel*: Variance in mean adaptation between wine yeasts, in each environment. *Bottom panel*: Variance in adaptation between replicated populations ($n = 24$ replicate ALE populations), for each wine yeast in each environment. Lineages mentioned in text are indicated (arrows). Source data are available online for this figure.

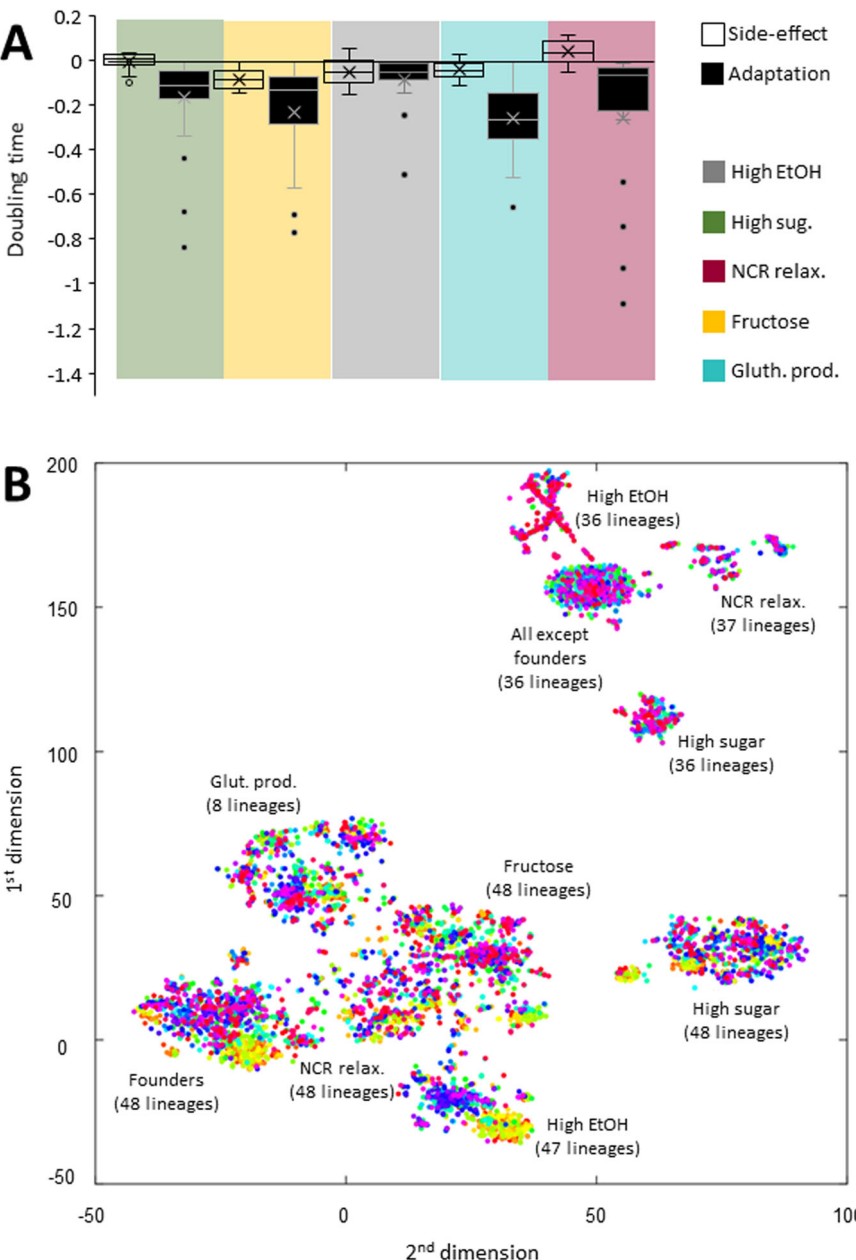

**Figure EV4. Side effects evolved by ALE wine yeasts.**

(A) Side effects (top box) of evolution are smaller than adaptations (bottom box), in all environments. Data across all side effects ($n = 18$ environments), and across all populations ($n = 24$ replicate ALE populations) of all lineages ($n = 48$ strains) in each ALE environment are shown. Box: interquartile range, line: median: whiskers 1.5× interquartile range, outliers: populations outside interquartile range. (B) t-Distributed Stochastic Neighbor Embedding (t-SNE) clustering reducing the variance in side effects to two dimensions (x, y-axes). Each dot represents one population of one lineage in one selection environment, or one starting population. The clustering is identical to that in Fig. 4, but color here indicates lineage. For large clusters, the selection regime and the number of lineages in the cluster are indicated. Note: side effects do not cluster by lineage and each cluster contains representatives of almost all lineages. Source data are available online for this figure.

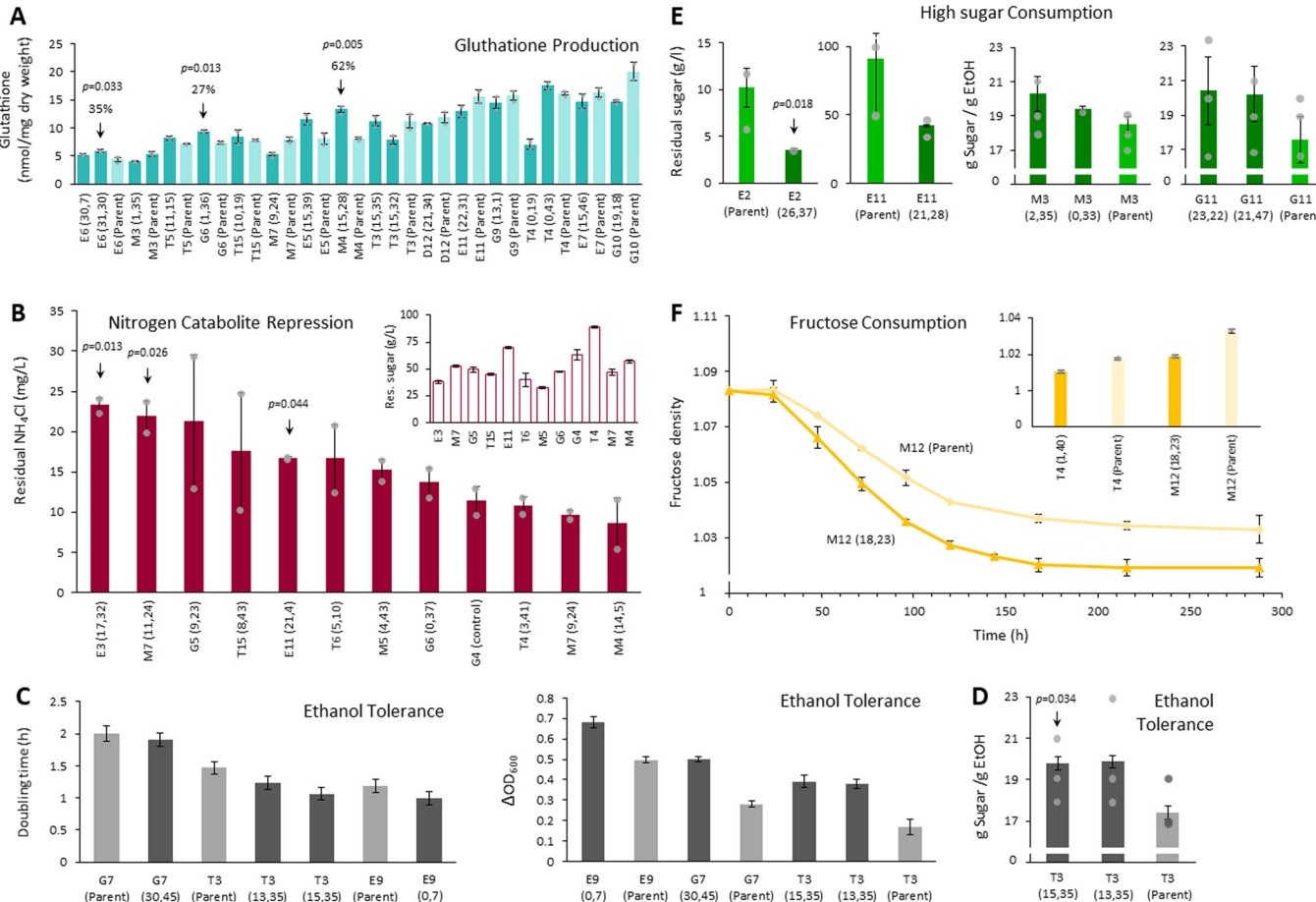

**Figure EV5. ALE wine yeasts retain adaptations in more industrial cultures.**

(A) Eighteen ALE populations selected for higher glutathione production. Arrows: populations producing more glutathione (one-sided Student's t test, $P < 0.05$) than their parental lineages in larger, liquid cultures. Total glutathione (nmol/mg dry weight biomass) at the end of a growth batch cycle (168 h) is shown. Error bars: SEM ($n = 2$ biological replicates). (B) Eleven ALE populations with improved growth in the NCR relaxation selection regime were cultivated in liquid synthetic grape must cultures (250 mL) with ammonium (43 mg N at start) as preferred nitrogen source and the residual ammonium was measured at the end of fermentation (360 h). The parental lineage G4, as the best-performing noncommercial wine strain, is shown as reference. Error bars: SEM ($n = 3$ biological replicates). Arrows indicate ALE populations selected for follow-up experiments. Note that non-ammonium nitrogen (total: 100 mg N at start) were present as arginine and proline. *Inset:* residual sugar bar plot (C) Six ALE populations, evolved for higher ethanol tolerance, and their parental lineages, were cultivated in low-volume liquid cultures (Bioscreen Inc.) in the presence of 8% ethanol. We tracked their growth continuously and extracted cell doubling times and cell yields. Error bars: SEM ($n = 3$ biological replicates). (D) We cultivated two ALE populations evolved for higher ethanol tolerance and expressing this trait in liquid cultures (see Fig. EV5C) in 50 mL synthetic grape must ($n = 2–3$ biological replicates) and compared their fermentation efficiency, measured as the gram sugar consumed per ethanol produced, to that of their parental lineage. Error bars: SEM. Arrow: one-sided Student's t test ($P < 0.05$) (E) 18 ALE populations, evolved for better growth in high-sugar concentrations, were cultivated in 40 mL liquid high-sugar grape must. We tracked the sugar consumption and ethanol production in each. Two ALE populations showed faster sugar uptake (*top panel*) than their parental lineages and four showed more efficient fermentation (*bottom panel*; gram ethanol produced per gram consumed). Error bars: SEM ($n = 3$ biological replicates) Arrow: significantly different (one-sided Student's t test, $P < 0.05$). (F) Top ALE populations T4 (1,40) and M12 (18,23), evolved for better fructose use, were cultivated in 40 mL liquid fructose containing synthetic grape must in the presence of the glucose analog 2-deoxyglucose. We compared their sugar fructose consumption to that of the parental lineages. Error bars: SEM ($n = 3$ biological replicates). *Inset:* end fructose density of two tested populations ($P = 0.05$). The line plot shows density kinetics for M12. Source data are available online for this figure.

