## [Peer Review File · Molecular Systems Biology]

Highly parallelized laboratory evolution of wine yeasts for enhanced metabolic phenotypes

Payam Ghiaci, Paula Jouhten, Nikolay Martyushenko, Helena Roca-Mesa, Jennifer Vazquez, Dimitrios Konstantinidis, Simon Stenberg, Sergej Andrejev, Kristina Grkovska, Albert Mas, Gemma Beltran, Eivind Almaas, Kiran Patil, and Jonas Warringer

Corresponding author(s): Kiran Patil (kp533@cam.ac.uk) , Jonas Warringer (jonas.warringer@cmb.gu.se), Eivind Almaas (eivind.almaas@ntnu.no)

Review Timeline:

Submission Date:	21st Mar 24
Editorial Decision:	22nd Mar 24
Revision Received:	3rd May 24
Editorial Decision:	16th May 24
Revision Received:	17th Jul 24
Accepted:	30th Jul 24

Editor: Maria Polychronidou / Poonam Bheda

Transaction Report: Please note that the manuscript was transferred from another journal where it was originally reviewed. Since the original reviews are not subject to EMBO's transparent review process policy, they cannot be published.

25th Mar 2024

Manuscript Number: MSB-2024-12326

Title: Highly parallelized laboratory evolution of wine yeasts for enhanced metabolic phenotypes

Dear Kiran,

Thank you again for submitting your work to Molecular Systems Biology. I have now read your manuscript, the referee reports from the other journal and your responses to these comments and discussed them with the team. Overall, we think that the review process has been constructive and therefore we can consider the study for publication using these reports and without reviewing the study from scratch. We think that most technical and other core concerns seem to have been addressed in the previous rounds of revision at the other journal. Reviewer #1 is, already from the first revision round, satisfied with the performed revisions and supports publication. Reviewer #2 lists a few remaining concerns, which mostly pertain to the final follow up experiments on the three selected strains in a 80L setup and whether they convincingly demonstrate the relevance of the approach for industrial applications. We think that the performed additional analyses and text edits that clarify the proof-of-concept nature of the 80L setup for indicating industrial applicability seem to address these remaining concerns of reviewer #2. Moreover, we note that both reviewers acknowledge the relevance of the study both for evolutionary biologists and for industrial microbiology. Taken together, we have decided to proceed with publishing the study in Molecular Systems Biology, pending some minor revisions related to editorial issues listed below.

- Please provide 5 keywords.
- The "one sentence summary" should be removed.
- Please include a Disclosure and Competing Interests Statement in the main text.
- Please note that our editorial policy does not allow "Data not shown".
- Please provide a .doc version of the manuscript text (including legends for main figures and tables) and individual files for the main figures. The figure legends should be included at the end of the manuscript text, after the References.
- The Reference list should be formatted according to the Molecular Systems Biology style i.e. ordered alphabetically and listing the first 10 authors followed by et al.
- We have replaced Supplementary Information by the Expanded View (EV format). In this case, all additional figures can be included in a PDF called Appendix. Appendix figures and Tables should be labeled and called out as: "Appendix Figure S1, Appendix Figure S2... Appendix Table S1..." etc. Each legend should be below the corresponding Figure/Table in the Appendix. Please include a Table of Contents in the beginning of the Appendix. For detailed instructions regarding expanded view please refer to our Author Guidelines: .
- We would encourage you to provide the data that are currently deposited at Mendeley Data as EV Datasets. Please provide one file per EV Dataset. Please include the description of each EV Dataset in the dataset file itself, ie. in a separate tab for .xls files or as a README.txt file in .zip folders. The EV Datasets should be called out in the text (even if it is only in the Data Availability section).
- Please provide a "standfirst text" summarizing the study in one or two sentences (approximately 250 characters), three to four "bullet points" highlighting the main findings and a "synopsis image" (550px width and max 400px height, jpeg format) to highlight the paper on our homepage.
- All Materials and Methods need to be described in the main text. We would encourage you to use 'Structured Methods', our new Materials and Methods format. According to this format, the Material and Methods section should include a Reagents and Tools Table (listing key reagents, experimental models, software and relevant equipment and including their sources and relevant identifiers) followed by a Methods and Protocols section in which we encourage the authors to describe their methods using a step-by-step protocol format with bullet points, to facilitate the adoption of the methodologies across labs. More information on how to adhere to this format as well as downloadable templates (.doc or .xls) for the Reagents and Tools Table can be found in our author guidelines: . An example of a Method paper with Structured Methods can be found here:
- Please include a Data availability section describing how the data, code etc. generated in this study have been made available. This section needs to be formatted according to the example below:
The datasets and computer code produced in this study are available in the following databases:
 - Chip-Seq data: Gene Expression Omnibus GSE46748 (<https://www.ncbi.nlm.nih.gov/geo/query/acc.cgi?acc=GSE46748>)

- [data type]: [full name of the resource] [accession number/identifier] ([doi or URL or identifiers.org/DATABASE:ACCESSION])
- Molecular Systems Biology supports formal data citations in the Reference list, to cite previously published datasets. In addition to citing the original papers that reported the data, we encourage you to also cite the relevant datasets directly in the Reference list. In the text, references to datasets are included as "Data ref: Smith et al, 2001" or "Data ref: NCBI Sequence Read Archive PRJNA342805, 2017". In the Reference list, data citations are very similar to normal literature references but must be labeled with "[DATASET]" at the end of the reference. For detailed instructions please refer to our Author Guidelines .
- When you resubmit your manuscript, please download our CHECKLIST (<https://bit.ly/EMBOPressAuthorChecklist>) and include the completed form in your submission. *Please note* that the Author Checklist will be published alongside the paper as part of the transparent process (<https://www.embopress.org/page/journal/17444292/authorguide#transparentprocess>)

Please resubmit your revised manuscript online, with a covering letter listing amendments and responses to each point raised by the referees. Please resubmit the paper ****within one month**** and ideally as soon as possible. If we do not receive the revised manuscript within this time period, the file might be closed and any subsequent resubmission would be treated as a new manuscript. Please use the Manuscript Number (above) in all correspondence.

Kind regards,

Maria

Maria Polychronidou, PhD
Senior Editor
Molecular Systems Biology

If you do choose to resubmit, please click on the link below to submit the revision online before 21st Apr 2024.

IMPORTANT:

- When assembling figures, please refer to our figure preparation guideline in order to ensure proper formatting and readability in print as well as on screen:
<https://bit.ly/EMBOPressFigurePreparationGuideline>
See also figure legend guidelines: <https://www.embopress.org/page/journal/17444292/authorguide#figureformat>

- Please note that corresponding authors are required to supply an ORCID ID for their name upon submission of a revised manuscript (EMBO Press signed a joint statement to encourage ORCID adoption).
(<https://www.embopress.org/page/journal/17444292/authorguide#editorialprocess>)
Currently, our records indicate that the ORCID for your account is 0000-0002-6166-8640.

Please click the link below to modify this ORCID:
Link Not Available

*** PLEASE NOTE *** As part of the EMBO Press transparent editorial process initiative (see our Editorial at <https://dx.doi.org/10.1038/msb.2010.72> , Molecular Systems Biology will publish online a Review Process File to accompany accepted manuscripts. When preparing your letter of response, please be aware that in the event of acceptance, your cover letter/point-by-point document will be included as part of this File, which will be available to the scientific community. More information about this initiative is available in our Instructions to Authors. If you have any questions about this initiative, please contact the editorial office (msb@embo.org).

Thank you for considering our submission MSB-2024-12326. Please find authors' (AU) responses to individual editorial comments (black) included below (red).

- Please provide 5 keywords.

AU: Five keywords are now included.

- The "one sentence summary" should be removed.

AU: The one sentence summary has been removed.

- Please include a Disclosure and Competing Interests Statement in the main text.

AU: We now declare no competing interests in the main text

- Please note that our editorial policy does not allow "Data not shown".

AU: All data referred to is shown.

- Please provide a .doc version of the manuscript text (including legends for main figures and tables) and individual files for the main figures. The figure legends should be included at the end of the manuscript text, after the References.

AU: The text is now included as .doc. Figure legends are included at the end of the text.

- The Reference list should be formatted according to the Molecular Systems Biology style i.e. ordered alphabetically and listing the first 10 authors followed by et al.

AU: The reference list is now formatted according to MSB style.

- We have replaced Supplementary Information by the Expanded View (EV format). In this case, all additional figures can be included in a PDF called Appendix. Appendix figures and Tables should be labeled and called out as: "Appendix Figure S1, Appendix Figure S2... Appendix Table S1..." etc. Each legend should be below the corresponding Figure/Table in the Appendix. Please include a Table of Contents in the beginning of the Appendix. For detailed instructions regarding expanded view please refer to our Author

Guidelines: <http://msb.embopress.org/authorguide#expandedview>.

AU: We have designated supplementary figures 1, 3, 4, 6 and 8 as EV figures 1-5, while supplementary figures 2, 5, 7, 9, 10 and 11 and all tables are now appendix.

- We would encourage you to provide the data that are currently deposited at Mendeley Data as EV Datasets. Please provide one file per EV Dataset. Please include the description of each EV Dataset in the dataset file itself, ie. in a separate tab for .xls files or as a README.txt file in .zip folders. The EV Datasets should be called out in the text (even if it is only in the Data Availability section).

AU: We have now included the Mendeley data as source data files in the submission, as instructed.

- Please provide a "standfirst text" summarizing the study in one or two sentences (approximately 250 characters), three to four "bullet points" highlighting the main findings and a "synopsis image" (550px width and max 400px height, jpeg format) to highlight the paper on our homepage.

AU: A standfirst text and a synopsis image is now included.

- All Materials and Methods need to be described in the main text. We would encourage you to use 'Structured Methods', our new Materials and Methods format. According to this format, the Material and Methods section should include a Reagents and Tools Table (listing key reagents, experimental models, software and relevant equipment and including their sources and relevant identifiers) followed by a Methods and Protocols section in which we encourage the authors to describe their methods using a step-by-step protocol format with bullet points, to facilitate the adoption of the methodologies across labs. More information on how to adhere to this format as well as downloadable templates (.doc or .xls) for the Reagents and Tools Table can be found in our author guidelines: <https://www.embopress.org/page/journal/17444292/authorguide#textformat>. An example of a Method paper with Structured Methods can be found here: <https://www.embopress.org/doi/10.15252/msb.20178071>.

AU: We have now added a Reagents and tools tables. However, we feel that a step-by-step protocol format with bullet points does not fit our Materials and Methods section well.

- Please include a Data availability section describing how the data, code etc. generated in this study have been made available. This section needs to be formatted according to the example below:

The datasets and computer code produced in this study are available in the following databases:

- Chip-Seq data: Gene Expression Omnibus GSE46748

(<https://www.ncbi.nlm.nih.gov/geo/query/acc.cgi?acc=GSE46748>)

- [data type]: [full name of the resource] [accession number/identifier] ([doi or URL or identifiers.org/DATABASE:ACCESSION])

AU: The data availability section is designed as instructed.

- Molecular Systems Biology supports formal data citations in the Reference list, to cite previously published datasets. In addition to citing the original papers that reported the data, we encourage you to also cite the relevant datasets directly in the Reference list. In the text, references to datasets are included as "Data ref: Smith et al, 2001" or "Data ref: NCBI Sequence Read Archive PRJNA342805, 2017". In the Reference list, data citations are very similar to normal literature references but must be labeled with "[DATASET]" at the end of the reference. For detailed instructions please refer to our Author Guidelines <http://msb.embopress.org/authorguide#datacitation>.

AU: We do not use or refer to any previously published data sets in this paper.

- When you resubmit your manuscript, please download our CHECKLIST (<https://bit.ly/EMBOPressAuthorChecklist>) and include the completed form in your submission. *Please note* that the Author Checklist will be published alongside the paper as part of the transparent process

<https://www.embopress.org/page/journal/17444292/authorguide#transparentprocess>)

AU: A check list has been included in the submission.

Please resubmit your revised manuscript online, with a covering letter listing amendments and responses to each point raised by the referees. Please resubmit the paper ****within one month**** and ideally as soon as possible. If we do not receive the revised manuscript within this time period, the file might be closed and any subsequent resubmission would be treated as a new manuscript. Please use the Manuscript Number (above) in all correspondence.

16th May 2024

Manuscript Number: MSB-2024-12326R

Title: Highly parallelized laboratory evolution of wine yeasts for enhanced metabolic phenotypes

Dear Kiran,

Thank you for sending us your revised manuscript. We have gone through the performed revisions and I am glad to inform you that we can soon accept the manuscript for publication, pending some final editorial requests listed below.

- Our data editors have noted the following issues that need to be fixed:

-- Please note that the figure 1 is missing in the manuscript, although the legend is provided. Figures 2-7 are mislabeled in the submissions system as figures 1-6 This needs to be corrected.

-- Please note that the legend for figure panels 2a-b is not labelled in the manuscript. Further, the legends 2c-d are mislabeled as 2a-b. This needs to be rectified.

-- Please indicate the exact p values in the legends of figures 7c; EV 5a, c, d-e.

-- Please include the information related to n in the legend of figure EV 3a.

-- Please note that n=2 in figure EV 5a. For n <5 we request that the actual individual data from each experiment should be plotted alongside an error bar.

-- Please describe the nature of entity for 'n' (biological? technical?) in the legends of figures 2c; 3c; 7a-b; EV 5a-f.

- The funding information provided in the manuscript text (Acknowledgements) should match the information entered in the online submission system. Currently the Research Council of Sweden grant no. 325-2014-6547 is missing from the submission system.

- The References should be formatted according to the Molecular Systems Biology reference style (i.e., ordered alphabetically and listing the first 10 authors followed by et al).

- The first reference in the References list seems to be misplaced, there is no mention of it in the text.

- Please correct the callouts for Appendix tables: i.e. for Appendix Tables S2 and S3 ("S" is missing from the callouts), Appendix Table S5 ("Appendix" is missing), Appendix Tables S6-S8 ("S" is missing and there isn't any Appendix Table S8), Appendix Figs. S1-S3 ("S" is missing). Moreover, the following callouts need to be corrected: Supplementary Fig. 1 and Supplementary Table 1 should be corrected to the respective EV or Appendix Figure/Table. Table EV8 is called out but doesn't exist.

- Table EV1A-D should be combined in a single Dataset EV1 file, with separate sheets and a description of the data in each sheet.

- Table EV2 should be provided as a single Excel file, with the description of the dataset in a separate sheet (instead of a .txt file in a .zip folder).

- Please include page numbers in the Appendix Table of Contents. Within the Appendix, Appendix Figure 1 needs "S" in the name.

Please resubmit your revised manuscript online, with a covering letter listing amendments and responses to each point raised by the referees. Please resubmit the paper ****within one month**** and ideally as soon as possible. If we do not receive the revised manuscript within this time period, the file might be closed and any subsequent resubmission would be treated as a new manuscript. Please use the Manuscript Number (above) in all correspondence.

Kind regards,

Maria

Maria Polychronidou, PhD

Senior Editor
Molecular Systems Biology

If you do choose to resubmit, please click on the link below to submit the revision online before 15th Jun 2024.

*** PLEASE NOTE *** As part of the EMBO Press transparent editorial process initiative (see our Editorial at <https://dx.doi.org/10.1038/msb.2010.72> , Molecular Systems Biology will publish online a Review Process File to accompany accepted manuscripts. When preparing your letter of response, please be aware that in the event of acceptance, your cover letter/point-by-point document will be included as part of this File, which will be available to the scientific community. More information about this initiative is available in our Instructions to Authors. If you have any questions about this initiative, please contact the editorial office (msb@embo.org).

Dear Kiran,

Thank you for sending us your revised manuscript. We have gone through the performed revisions and I am glad to inform you that we can soon accept the manuscript for publication, pending some final editorial requests listed below.

- Our data editors have noted the following issues that need to be fixed:

AU: Comments are included in red below

-- Please note that the figure 1 is missing in the manuscript, although the legend is provided. Figures 2-7 are mislabeled in the submissions system as figures 1-6 This needs to be corrected.

AU: We apologize. Figure 2 had failed to export correctly. The figure is now included and subsequent figures re-labelled. Figure 1 is identical to the synopsis figure. We have now included both in the submission.

-- Please note that the legend for figure panels 2a-b is not labelled in the manuscript. Further, the legends 2c-d are mislabeled as 2a-b. This needs to be rectified.

AU: The legend for figure 2 is now correct. The mismatch of legend to figure followed from Figure 2 not exporting correctly.

-- Please indicate the exact p values in the legends of figures 7c; EV 5a, c, d-e.

AU: We have now added the exact p-values to the figures 7c, EV 5a, b, d and e.

-- Please include the information related to n in the legend of figure EV 3a.

AU: This is now reported as: "mean of (\log_2 , normalized) cell doubling time for each wine yeast ($n=24$ replicate ALE populations, extinct populations excluded, in each environment) across all environments ($n=8$ environments). Error bars: SEM ($n=8$)."

-- Please note that $n=2$ in figure EV 5a. For $n < 5$ we request that the actual individual data from each experiment should be plotted alongside an error bar.

AU: We have now added individual data points for figure EV5 b, d, and e. For Figure EV5A, measurement errors are too small for individual data points to be distinguishable. We note that individual data points for all figures are reported in the supplementary data.

-- Please describe the nature of entity for 'n' (biological? technical?) in the legends of figures 2c; 3c; 7a-b; EV 5a-f.

AU: We have multiple layers of biological replication. We now distinguish between these in all legends as "strains", "replicate ALE populations" and "biological replicates" respectively, in all instances where "n" is mentioned. There are no technical replicates in the study.

- The funding information provided in the manuscript text (Acknowledgements) should match the information entered in the online submission system. Currently the Research Council of Sweden grant no. 325-2014-6547 is missing from the submission system.

AU: We have now added the grant to the online submission system.

- The References should be formatted according to the Molecular Systems Biology reference style (i.e., ordered alphabetically and listing the first 10 authors followed by et al).

AU: The references have now been formatted correctly.

- The first reference in the References list seems to be misplaced, there is no mention of it in the text.

AU: The first reference has been removed.

- Please correct the callouts for Appendix tables: i.e. for Appendix Tables S2 and S3 ("S" is missing from the callouts), Appendix Table S5 ("Appendix" is missing), Appendix Tables S6-S8 ("S" is missing and there isn't any Appendix Table S8), Appendix Figs. S1-S3 ("S" is missing). Moreover, the following callouts need to be corrected: Supplementary Fig. 1 and Supplementary Table 1 should be corrected to the respective EV or Appendix Figure/Table. Table EV8 is called out but doesn't exist.

AU: Appendix Tables are now correctly called out. Supplementary Fig. 1 and Fig S1 is now correctly called out as Fig EV1. Supplementary Table 1 is now correctly referred to as Appendix Table S1. Appendix Table S7 is now correctly called out as Table EV1. Tables EV8 and Appendix Table S8 are now correctly referred to as EV2. We have also changed one instance of Fig S8C to Fig EV5C.

- Table EV1A-D should be combined in a single Dataset EV1 file, with separate sheets and a description of the data in each sheet.

AU: Done. Correspondingly, we now refer to table EV1A-D as table EV1

- Table EV2 should be provided as a single Excel file, with the description of the dataset in a separate sheet (instead of a .txt file in a .zip folder).

AU: Done.

- Please include page numbers in the Appendix Table of Contents. Within the Appendix, Appendix Figure 1 needs "S" in the name.

AU: Page numbers are included in the Appendix and referred to in the appendix table of contents. Appendix Figure S1 is now correctly labelled.

30th Jul 2024

Manuscript number: MSB-2024-12326RR

Title: Highly parallelized laboratory evolution of wine yeasts for enhanced metabolic phenotypes

Dear Prof Patil,

Thank you again for sending us your revised manuscript. We are now satisfied with the modifications made and I am pleased to inform you that your paper has been accepted for publication.

Yours sincerely,

Poonam Bheda, PhD
Scientific Editor
Molecular Systems Biology
